**EMBO** *reports*

# Transcriptome-wide decoding the roles of aberrant splicing in melanoma MAPK-targeted resistance evolution

Jing Yu[1,7], Xiujing He[1,7], Xueyan Wang[1,7], Chune Yu[2,7], Xian Jiang[1], Yanna Li[1], Xinyu Liu[1], Ya Luo[3], Xuemei Chen[4], Sisi Wu [4], Lu Si [5✉], Jing Jing [1✉], Xuelei Ma [6✉] & Hubing Shi [1✉]

## Abstract

**Drug resistance critically limits the long-term efficacy of MAPK-targeted therapy in melanoma. While resistance mechanisms at genetic, epigenetic, and transcriptional scales are well-documented, post-transcriptional splicing regulation remains poorly understood. By analyzing patient-matched pre-treatment and resistant melanoma biopsies, we uncover widespread alternative splicing alterations during therapy resistance. Splicing perturbations are most pronounced in MAPK and PI3K-AKT pathway genes. We identify a splicing switch of AKT2 from isoform 210 to 206 in 29.55% (13/44) of disease-progressive biopsies. This splicing switch induces AKT2 kinase hyperactivity by restoring the activated fragment A-loop. Functional validations confirm that AKT2-206 confers BRAF inhibitor resistance in melanoma cells by activating S6 kinase. Further, the splicing factor hnRNPK likely drives the splicing switch of AKT2 during acquired resistance. Our results not only provide insights into splicing-mediated regulation of drug resistance but also highlight the importance of alternative splicing isoforms as targets for clinical diagnosis and therapy.**

**Keywords** Drug Resistance; Alternative Splicing; AKT2; hnRNPK; Melanoma
**Subject Categories** Cancer; RNA Biology

## Introduction

BRAF inhibitors (BRAFi), whether used as monotherapy or in combination with MEK inhibitors (MEKi), have demonstrated remarkable clinical efficacy in a substantial proportion of cancer patients, particularly those with melanoma harboring mutant

$BRAF^{V600E}$. However, acquired drug resistance poses a formidable obstacle to achieving long-term benefits from these treatment regimens. Multiple mechanisms of acquired resistance hitherto unraveled fall into two categories, reactivation of the mitogen-activated protein kinase (MAPK) pathway and activation of alternative compensatory pathways. Aberrations in the reactivated MAPK pathway, such as genetic alterations of *RAS* (Nazarian et al, 2010) and *MEK* (Emery et al, 2009), *BRAF* amplification (Shi et al, 2012), loss of *NF1* (Whittaker et al, 2013), deregulation of *COT1* (Johannessen et al, 2010), and *BRAF* alternative splicing (Poulikakos et al, 2011), are frequently detected in melanoma progression despite treatment (Shi et al, 2014b). The PI3K-AKT pathway, as a prominent alternative pathway, was found to be upregulated in 22% of disease-progressive melanomas, eliciting tumor escape from BRAFi and MEKi therapies (Shi et al, 2014b). Among these resistant melanomas, *BRAF* splicing is one of the most frequently occurring driving events, with a recurrence rate of 31.58% (Poulikakos et al, 2011), underscoring the potential contribution of alternative splicing to drug tolerance.

Over 90% of human genes undergo alternative splicing following transcription, a process that expands genetic diversity beyond the apparently limited number of genes (Wang et al, 2008). Alternative splicing is sophisticatedly controlled by multilayered regulatory processes, inducing vast variations in both coding sequences and noncoding regions. These variations may impact mRNA stability, subcellular localization, and translational efficiency, thereby resulting in the generation of diverse protein isoforms with varied functions and/or localizations (Baralle and Giudice, 2017). Moreover, alternative splicing also affects functional domains within protein families commonly mutated in tumors and potentially remodels protein–protein interactions of cancer driver genes (Climente-González et al, 2017). Aberrant splicing events are frequently implicated in tumor progression, metastasis, and drug resistance, thus emerging as promising therapeutic targets. The function of aberrant splicing in promoting resistance to targeted therapy has been highlighted, involving the oncogenes *BCR-ABL*,

[1]Institute of Breast Health Medicine, State Key Laboratory of Biotherapy, West China Hospital, Sichuan University and Collaborative Innovation Center, 610041 Chengdu, Sichuan, China. [2]State Key Laboratory of Systems Medicine for Cancer, Shanghai Cancer Institute, Renji Hospital, Shanghai Jiao Tong University School of Medicine, 200032 Shanghai, China. [3]Department of Blood Transfusion, Laboratory Medicine Center, The Second Affiliated Hospital, Army Military Medical University, 400037 Chongqing, China. [4]Core Facilities of West China Hospital, Sichuan University, 610041 Chengdu, Sichuan, China. [5]Key Laboratory of Carcinogenesis and Translational Research (Ministry of Education/Beijing), Department of Renal Cancer and Melanoma, Peking University Cancer Hospital & Institute, 100084 Beijing, China. [6]Department of Biotherapy, West China Hospital and State Key Laboratory of Biotherapy, Sichuan University, 610041 Chengdu, Sichuan, China. [7]These authors contributed equally: Jing Yu, Xiujing He, Xueyan Wang, Chune Yu. ✉E-mail: silu15_silu@126.com; jingjing@wchscu.edu.cn; drmaxuelei@gmail.com; shihb@scu.edu.cn

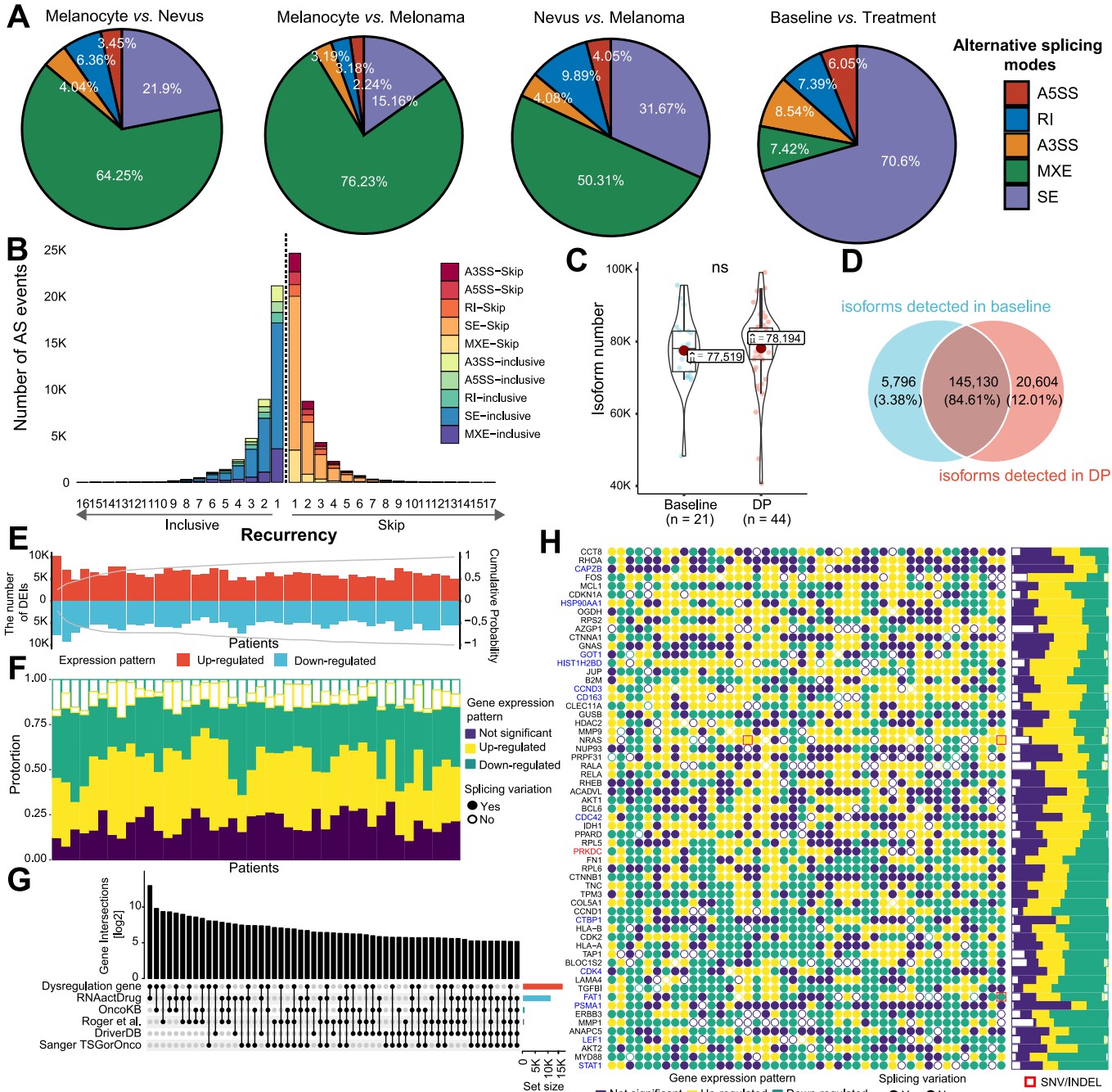

BIM, IKZF1, BRCA1, TP53, BRAF, CD19, AR, ER, and PIK3CD (Singh and Eyras, 2016). In particular, alternative splicing of $BRAF^{V600E}$ caused selective elimination of the RAS-binding domain and continuous activation in a RAS-independent manner, thereby facilitating the acquisition of drug resistance in melanoma (Poulikakos et al, 2011). Moreover, alternative splicing isoforms of non-mutated NRAS also confer BRAFi resistance via activation of the MAPK and PI3K pathways (Eisfeld et al, 2014). Furthermore, in a comprehensive study of multiple cancer types, alternative splicing and somatic mutations, which impact similar functions in tumors, often show an exclusive regulatory manner (Climente-González et al, 2017). This strongly suggests that specific alterations in splicing could become prospective therapeutic targets, especially for tumors with a low mutation burden in driver oncogenes. These

findings collectively highlight the significance of reprogramming alternative splicing in the development of acquired drug resistance.

Given this, we established a high-throughput approach to systematically dissect splicing alterations during the evolution of acquired drug resistance. Alternative splicing isoforms in BRAFi-relapsed melanoma were compared with those in patient-matched baseline biopsies. The profiling of functional splicing isoforms potentially driving the acquisition of drug resistance was depicted. A splicing switch in AKT2 was identified and functionally validated as a potential contributor to the development of drug resistance. Heterogeneous nuclear ribonucleoprotein K (hnRNPK), an RNA-binding protein potentially regulating the alternative splicing of AKT2, was also identified through correlation analysis and subsequent functional validation. Thus, our results offer insights

**Figure 1. Comprehensive analysis of splicing alterations during the evolution of MAPKi resistance.**

(A) Characteristics of differential splicing events at various stages of melanoma development and drug resistance evolution. The pie charts illustrate the distribution of various splicing event modes. Different colors indicate distinct modes of splicing events. (B) Recurrence of each identified differential splicing event in distinct sample pairs between the Baseline and DP groups. (C) Comparison of the splicing isoform counts between Baseline and DP groups. Boxes show median (center line) and 25–75th percentiles (bounds), whiskers indicate 1.5 × interquartile range, individual data points are overlaid. The red dots denote the mean values. ns indicates $P$ value > 0.05; Student's $t$ test. (D) Venn analysis of splicing isoforms identified within Baseline and DP groups. (E) The number of upregulated and downregulated isoforms identified in each matched sample pair. (F) The histogram shows the consistency of expression changes at the isoform and gene levels. The color of the histogram represents the gene expression change status: purple denotes no significant change, yellow denotes significant upregulation, and green represents significant downregulation. The empty histogram signifies an insignificant isoform expression change, while a solid histogram indicates a significant change. Specifically, a solid yellow histogram signifies genes significantly upregulated at both the gene and transcript levels. (G) Crossover analysis between the genes we identified that were significantly changed at the isoform level and other tumor-related gene sets. The drug sensitivity gene set from RNAactDrug database encompasses 11,308 genes associated with MAPKi sensitivity; the OncoKB cancer gene set comprises 1064 oncogenes; the Roger cancer gene set includes 855 genes associated with cancer, melanoma, MAPKi drug resistance, and immunotherapy; the DriverDBv3 SKCM gene set consists of 317 melanoma driver genes; and the Sanger TSGorOnco gene set contains 279 tumor suppressor genes and oncogenes. (H) Expression alterations of BRAFi resistance-related genes at both the gene and isoform levels throughout the progression of MAPKi resistance development. The circle on the left illustrates the alterations in gene expression and alternative splicing of 60 melanoma BRAFi resistance-related genes. The top 30 corresponding genes were ranked and selected based on the reproducibility of isoform upregulation and downregulation across each sample pair. The histogram on the right displays the expression changes at the gene and isoform levels, as illustrated in (F). Oncogenes are highlighted in red, and tumor suppressor genes are highlighted in blue, as defined by the Roger cancer gene set mentioned previously. Genes exhibiting SNP/INDEL variation are indicated by red boxes. SE skipped exon, A5SS alternative 5′ splice site, A3SS alternative 3′ splice site, MXE mutually exclusive exons, RI retained intron, A3SS-Skip skipping of the proximal 3′ splice site, A5SS-Skip skipping of the distal 5′ splice site, MXE-Skip skipping of the upstream exon, A3SS-inclusive inclusion of the proximal 3′ splice site, A5SS-inclusive inclusion of the distal 5′ splice site, MXE-inclusive inclusion of the upstream exon. Baseline, biopsy samples before MAPK-targeted therapy; DP, biopsy samples resistant to MAPK-targeted therapy. Source data are available online for this figure.

## Results

### Genome-wide splicing alterations in MAPKi-resistant evolution of melanoma

To provide an overview of alternative splicing changes throughout melanoma oncogenesis and the progression of acquired drug resistance, we reanalyzed transcriptome profiles of melanocyte, nevus, melanoma, and melanoma treated with MAPK inhibitors (GSE138711 (Hanniford et al, 2020a), GSE98394 (Badal et al, 2017a), GSE65186 (Hugo et al, 2015a), and EGAS00001000992 (Kwong et al, 2015)). We first characterized five canonical alternative splicing patterns by rMATS, namely exon skipping/inclusion (SE), alternative 5′ splice-site selection (A5SS), alternative 3′ splice-site selection (A3SS), intron retention (RI), and mutually exclusive exons (MXE). The proportions of differential alternative splicing modes of melanocyte vs. nevus, melanocyte vs. melanoma, and nevus vs. melanoma were similar (Fig. 1A). However, a unique differential splicing pattern was observed between baseline and disease-progressive (DP) tumors treated with MAPK-targeted therapy, with SE being the most predominant splicing mode (Fig. 1A,B). Genes harboring differential splicing events in the DP comparison group were strongly enriched in signal transduction pathways associated with MAPKi resistance, including mTOR and PI3K-AKT signaling pathways (Appendix Fig. S1). This result highlights a specific process of alternative splicing during the development of drug resistance. To thoroughly investigate this process, we profiled the dynamic alterations of splicing isoforms during resistance evolution using three independent algorithms, namely Cufflinks (Trapnell et al, 2012), StringTie (Pertea et al, 2016), and AIDE (Li et al, 2019). We present the results processed by AIDE, as it has a favorable balance of

accuracy and sensitivity in isoform identification (Appendix Fig. S2). A total of 171,530 unique isoforms were identified across all biopsies, corresponding to 30,555 genes. The isoform heterogeneity of genes harboring differential splicing events (identified by rMATS) is significantly higher than that of non-altered genes (Appendix Fig. S3). No significant difference in isoform counts was observed between DP biopsies ($n = 44$) and baseline biopsies ($n = 21$, Fig. 1C). Venn diagram further showed that these two types of biopsies shared as high as 84.61% of the isoforms (Fig. 1D). At the expression level, 44,966 differential expression isoforms (DEIs), representing 27.31% of human genes, were identified during disease progression (Dataset EV1). The numbers of upregulated and downregulated isoforms in each patient-matched biopsy pair were comparable, implying that disease progression was accompanied by a selection of splicing isoforms rather than an overall promotion or suppression of mRNA splicing (Fig. 1E). The cumulative curve of upregulated and downregulated isoforms indicated a high recurrence of the DEIs in this cohort (Fig. 1E).

Subsequently, we incorporated the splicing alteration data and gene expression data (Fig. 1F). We found that alternative splicing added further regulatory diversity beyond gene expression variation. Notably, splicing alterations were observed in an average of 20.44% of genes without expression variation (Fig. 1F). Previous investigations at the transcriptome level overlooked the contribution of these alterations to disease progression. To evaluate whether these DEIs are involved in drug resistance, we investigated the overlap between DEIs and oncogenes from public databases. The results showed that these DEIs are related to oncogenesis and drug sensitivity (Fig. 1G). According to a previous study, these intrinsic oncogenic genes also play a consistent role in the development of drug resistance (Nazarian et al, 2010). We then checked BRAFi resistance-related genes that were comprehensively summarized in our previous study (Hugo et al, 2015b). Notably, 76.73% (656 of 855) of cancer-, melanoma-, and immune-related genes exhibited aberrant splicing switches in one or more biopsies during the acquisition of drug resistance (Fig. 1H; Dataset EV1).

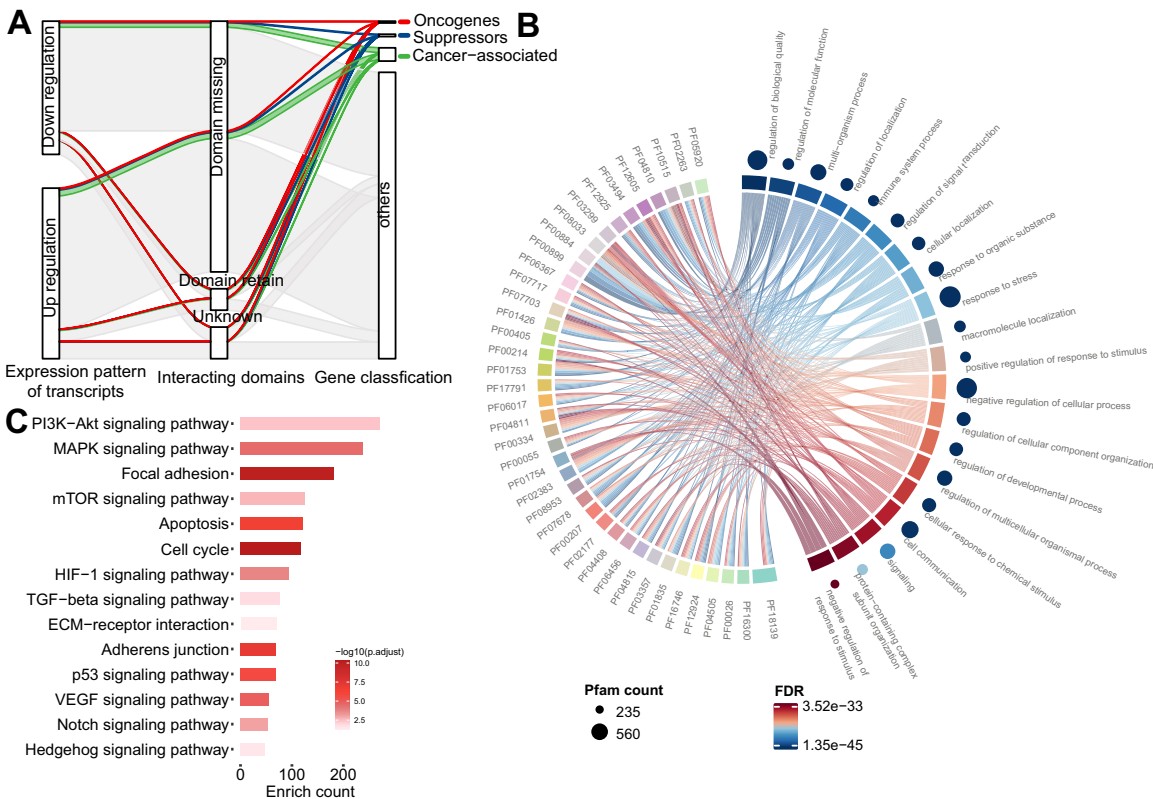

**Figure 2. Functional annotation of differentially expressed splicing isoforms.**

(A) Domain changes caused by alternative splicing. The protein domains in DEIs were annotated. Oncogenes, suppressors, and other cancer-associated genes are highlighted in different colors. (B) Gene Ontology enrichment analysis of differentially expressed splicing isoforms. Pfam conserved domains affected by alternative splicing were clustered into Gene Ontology categories using domain-centric Gene Ontology (dcGO). The color and size of points in the chords represent the False Discovery Rate (FDR) of enriched biological processes and the number of Pfam protein domains, respectively, linking each Pfam to the processes it is associated with. Only the top 20 significant GO terms (FDR ≤0.05) and the top 40 Pfam terms with high recurrence are shown. (C) Pathway enrichment analysis of differentially expressed splicing isoforms. The y axis represents the pathway, and the x axis shows the number of genes with splicing alteration. The color of the bar represents the adjusted P value. Hypergeometric test. Source data are available online for this figure.

## Identification of functional splicing isoforms related to resistance to MAPK-targeted therapy

The complicated and heterogeneous expression patterns of splicing isoforms pose considerable obstacles to identifying functional splicing isoforms associated with drug resistance. To tackle this problem, we designed a rational data mining flowchart (Appendix Fig. S4). First, coding isoforms were extracted from DEIs and ranked based on their occurrence ratios during disease progression. Approximately 56.30% of the isoforms (25,318 out of 44,966) were obtained. Second, alternative splicing isoforms encoding functional protein domains were pulled out. Surprisingly, the majority of these splicing alterations (82.54%, 20,898 out of 25,318) impaired conserved protein domains, potentially leading to gain or loss of function (Fig. 2A). Then, Gene Ontology (GO) enrichment analysis was conducted on these isoforms with disrupted domains to assess their associated biological processes. The results showed that these domains or isoforms were enriched in 72 biological processes and were predominantly correlated with signaling pathways (Fig. 2B; Table EV1). Finally, to determine the predominant pathways involved in MAPKi resistance, we performed an enrichment analysis of cancer-related pathways. We found that the PI3K-

AKT and the MAPK pathways were the most relevant pathways, highlighting dramatic changes in alternative splicing within these pathways (Fig. 2C).

## Alternative splicing in the MAPK pathway confers MAPKi resistance

According to a previous study, reactivation of the MAPK pathway caused by *BRAF* and *NRAS* alternative splicing conferred BRAFi resistance in melanoma cells (Emery et al, 2009). Therefore, we first investigated the roles of splicing isoforms within the MAPK pathway in regulating drug resistance. We projected altered splicing isoforms to the modules of the MAPK pathway based on the KEGG module database. Up to 63 splicing isoforms in four MAPK modules were identified with distinct expression patterns during MAPKi resistance evolution (Appendix Fig. S5), of which 15.87% showed recurrent deregulation in the Skin Cutaneous Melanoma (SKCM) cohort from The Cancer Genome Atlas (TCGA) (Table EV2). Consistent with previous reports (Duggan et al, 2017; Poulikakos Poulikos I et al, 2011), significant splicing alterations were identified in kinases of the canonical MAPK cascade, including the oncogenes *BRAF*, *NRAS*, and *KRAS*, which might

potentially contribute to MAPKi resistance in melanoma (Appendix Figs. S5 and S6).

Furthermore, a series of genes in non-canonical MAPK modules, including p38, JNK, and ERK5, also presented extensive splicing alterations (Appendix Fig. S5A). This suggested that these alternative isoforms may also contribute to MAPKi-acquired resistance. To validate the function of the identified isoforms, we overexpressed CD14-203, MAP2K3-203, and RAP1B-229 in MAPKi-sensitive cell lines. The results of short- and long-term cell proliferation assays showed that CD14-203, but not MAP2K3-203 or RAP1B-229, conferred BRAFi resistance (Appendix Figs. S7 and S8). This finding implies that only a few splicing isoforms play functional driver roles. Whereas, the remaining isoforms may undergo passive regulation and processing by the spliceosome during the development of MAPKi resistance.

## The splicing isoform of AKT2 as a potential driver of drug resistance development

As shown in Fig. 2C, in addition to the MAPK pathway, the PI3K-AKT pathway was another enriched predominant signaling. Thus, we focused on the altered splicing isoforms within this pathway. Up to 55 splicing isoforms representing 43 genes were differentially expressed during the evolution of drug resistance (Table EV3). RNAactDrug database revealed 31 genes functionally tied to drug sensitivity (Dong et al, 2020), while 11 splicing isoforms corresponding to 10 genes also presented aberrant expression in SKCM cohort from TCGA, implicating these genes in disease progression. Collectively, these findings highlight a possible dual biological mechanism for the 43 genes identified above, which may be linked to both drug resistance-driven adaptation and tumorigenesis.

Among these functionally altered isoforms, we found a significant downregulation of an AKT2 isoform, AKT2-210 (Fig. 3A). As reported by our group and others, the mutation and upregulation of AKT1/3 potently confer drug resistance to BRAFi or BRAFi plus MEKi (Shi et al, 2014a; Shi et al, 2014b). Moreover, AKT3 also plays a pivotal role in melanoma tumorigenesis and drug resistance (Shao and Aplin, 2010). These observations led us to hypothesize that alternative splicing of AKT2 may represent an additional mechanism contributing to MAPKi resistance, akin to other members of the AKT family. We profiled both annotated and novel AKT2 isoforms, which were significantly altered in drug-resistant biopsies. AKT2-210 and AKT2-206 were the most significantly downregulated and upregulated isoforms, respectively (Fig. 3B–D). AKT2-210 was recurrently downregulated in 22 out of 44 (50.00%) DP biopsies, whereas AKT2-206 upregulation was frequently observed in 15 out of 44 DP biopsies (34.09%, Fig. 3E). Consistent with these findings, transcriptome sequencing data from five independent studies (GSE75299, GSE203545, GSE285131, GSE103630, GSE186108), showed that AKT2-206 expression was significantly increased in 57.7% (15/26) of acquired resistance samples (Appendix Fig. S9A). Notably, a negative correlation between AKT2-210 and AKT2-206 was observed within patient-matched biopsy pairs ($R = -0.562$, $P$ value < 0.001; Fig. 3F). This reciprocal splicing switch may explain why no significant change was observed at the gene expression level of AKT2 between baseline and disease-progressive biopsies in our previous work (Hugo et al, 2015b).

Next, we dissected the biological domain information of the identified AKT2 isoforms. The results revealed that the splicing switch between AKT2-210 and AKT2-206 primarily impacted the integrity of the kinase domain PKB (Fig. 4A). Structural comparisons revealed that the PKB domain of AKT2-210 was slightly shorter than that of AKT2-206, which is necessary for full kinase function (Fig. 4A). Blasting the sequences of AKT2-210 and AKT2-206, we found that exon 10 was absent in AKT2-210. Consistent with this observation, we found that there was a significant correlation between the log2FC value of AKT2-206 and the Delta-Percent-Spliced-In (PSI) value of exon 10 ($R = 0.41$, $P = 0.038$, Appendix Fig. S9B), indicating that changes in AKT2-206 expression may be associated with the splicing regulation of exon 10. Alignment of AKT family variants across multiple species revealed that the amino acid sequence encoded by this exon was highly conserved during biological evolution (Fig. 4A). The skipping of exon 10 in AKT2 results in the deletion of a 43-amino acid peptide in-frame (Fig. 4A). Importantly, the activated loop (A-loop), which is a functional segment of the AKT family, was also included in this peptide. Structure modeling predicted that the loss of the A-loop, encompassing the regulatory residue Thr 309 and catalytic residue Arg 274, disrupted the interaction and catalysis of the substrate ATP (Fig. 4B). Collectively, the splicing switch from AKT2-210 to AKT2-206 may potentially activate the AKT downstream pathway, thereby conferring MAPKi resistance.

## AKT2-206 confers MAPKi resistance by activating the PI3K-AKT pathway

Given the pivotal role of AKT in both the MAPK and PI3K-AKT signaling pathways, we validated the biological function of the identified AKT2 isoforms regarding conferring MAPKi resistance in melanoma. SK-MEL-28 and A375 cells, with induced ectopic expression of AKT2-210 and AKT2-206, were subjected to the anti-apoptosis assay in the presence of the BRAF inhibitor PLX4032 (Fig. 5). The results of the apoptosis assay demonstrated that the functional isoform AKT2-206 rather than AKT2-210 prevented both early and late apoptosis in the presence of PLX4032 (Fig. 5A). Similar to the apoptosis assay results, the long-term and short-term proliferation assays showed that the overexpression of AKT2-206 but not of AKT2-210 or the blank vector potently promoted BRAFi tolerance in SK-MEL-28 cells (Appendix Fig. S10). To explore therapeutic potential, we designed an antisense oligonucleotide (ASO) targeting exon 10. ASO-mediated silencing of exon 10 significantly increased the sensitivity of melanoma cells to PLX4032, suggesting that targeting AKT2 splicing might represent a strategy to overcome MAPKi resistance (Appendix Fig. S11). As the PI3K-AKT signaling pathway regulates anti-apoptotic activity mediated by AKT phosphorylation, we observed hyperphosphorylation of AKT in model cells overexpressing AKT2-206 but not in those overexpressing AKT2-210 or the blank vector (Fig. 5B). Furthermore, AKT2-210 over-expression cells showed increased sensitivity to both pan-AKT inhibitors (pan-AKTi, capivasertib) and AKT2-specific inhibitors (AKT2i, CCT128930) compared to AKT2-206 overexpression cells (Appendix Fig. S12A). Notably, activation of AKT was independent of the MAPK cascades MEK1/2 or ERK1/2 for up to 24 h (Appendix Fig. S12B), indicating that acquired drug resistance is primarily attributed to the PI3K-AKT pathway.

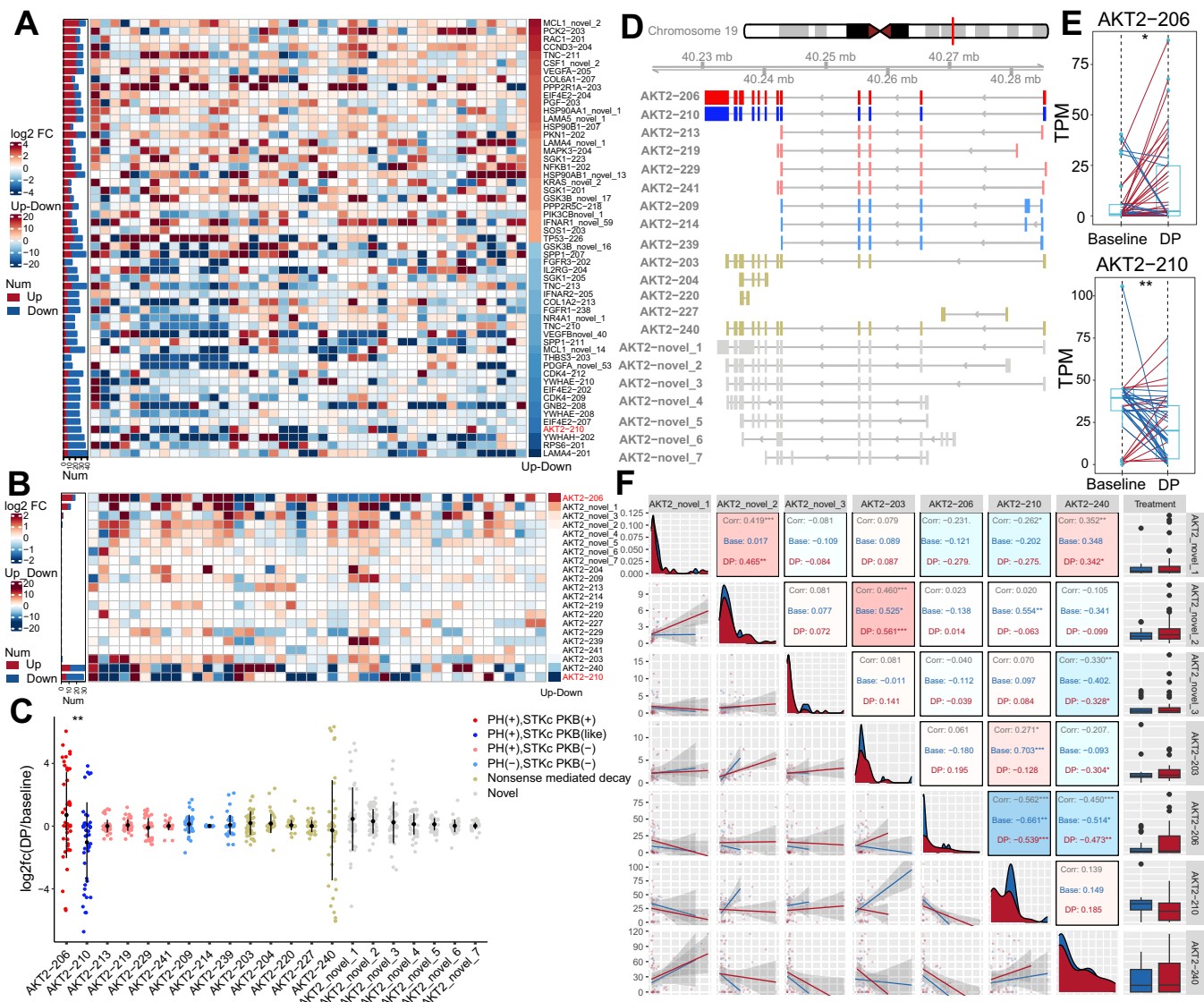

**Figure 3. MAPKi triggers the splicing switch of AKT2 in melanoma.**

(A) MAPKi induces extensive splicing changes in the PI3K-AKT pathway. Splicing isoforms were ranked based on the difference between the number of upregulation and downregulation events, and the corresponding color bar is on the right. The stacked bar chart on the left panel shows the upregulation (red) and downregulation (blue) events of each splicing isoform. (B) Splicing switch of AKT2 during the acquisition of MAPKi resistance. (C) Change patterns of all detected AKT2 splicing isoforms during drug resistance evolution. AKT2 splicing isoforms are marked in distinct colors according to the presence of PH domain and PKB domain. Novel splicing isoforms identified in this study are displayed in gray. The data are presented as mean ± SD ($n = 44$, biological replicates). Student's $t$ test. (D) Exon structure and splice sites of AKT2 splicing isoforms. The color code is the same as in (C). (E) Expression patterns of AKT2-206 and AKT2-210 in patient-matched pairs. Upregulation and downregulation patterns are displayed in red and blue, respectively. Boxes show median (center line) and 25–75th percentiles (bounds), whiskers indicate 1.5 × interquartile range, dots represent each sample ($n = 44$, biological replicates). Paired $t$ test, AKT2-206 ($P = 0.019$), AKT2-210 ($P = 0.0042$). (F) Correlation analysis of AKT2 splicing isoforms with expression changes during MAPK-targeted therapy. Pearson's correlation coefficient is shown above the diagonal (top right). Boxes show median (center line) and 25–75th percentiles (bounds), whiskers indicate 1.5 × interquartile range, dots are outliers (base = 21, DP = 44, biological replicates). *$P < 0.05$, **$P < 0.01$, ***$P < 0.001$, or ****$P < 0.0001$. Source data are available online for this figure.

Encouraged by the in vitro findings, we subsequently validated the role of AKT2-206 in conferring resistance to MAPKi using a mouse model xenografted with SK-MEL-28 cells transfected with vector, AKT2-210, and AKT2-206. Although PLX4032 significantly decelerated tumor growth, the overexpression of AKT2-206, rather than AKT2-210, reversed the growth inhibition (Fig. 5C–E). In line with the tumor growth curves, immunohistochemical staining of

Ki67, a proliferation marker, demonstrated the growth-promoting effect of AKT2-206 in the presence of PLX4032 (Fig. 5F). Furthermore, the phosphorylation levels of AKT and downstream S6 kinase remained relatively high in AKT2-206 overexpression tumors, irrespective of PLX4032 treatment (Fig. 5F). Taken together, these results indicate that AKT2-206 confers MAPKi resistance by activating the PI3K-AKT pathway.

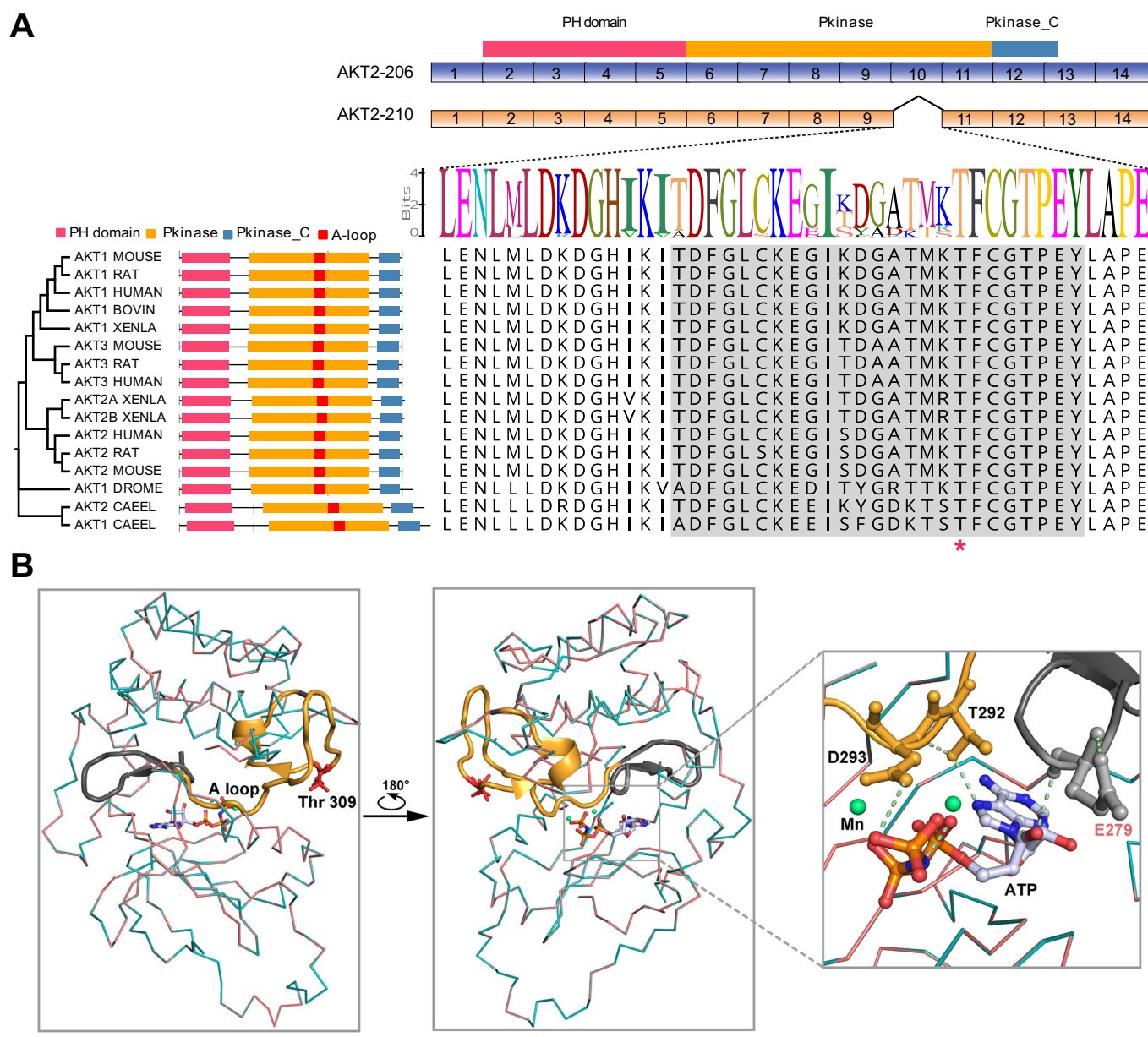

**Figure 4. Schematic representation of alterations in AKT2 splicing isoforms.**

(A) Exon composition of AKT2-206 and AKT2-210. The skipping of exon 10 in AKT2 results in the absence of the A-loop. The A-loop is highlighted in gray block, and the regulatory residue Thr 309 is marked with asterisks. (B) Superimposition of AKT2-206 (blue) and AKT2-210 (red), with an ATP analog (AMP-PNP) shown as sticks. The regulatory residue, Thr 309 of the activation segment, is shown as sticks in red. A loop is marked in yellow. The altered region is highlighted in light gray. The green sphere indicates the location of $Mn^{2+}$. A zoomed-in view of the interface between the A-loop of AKT2 and ATP suggests that the altered region likely abolishes a critical interaction with ATP. Direct hydrogen bonds between AKT2 and ATP are shown as dashed lines. The box indicates the regions shown in zoom-in. Source data are available online for this figure.

## hnRNPK activates the PI3K-AKT pathway by mediating the splicing switch of AKT2

We then sought to elucidate the regulatory mechanisms that perturb BRAFi sensitivity in tumor cells by controlling the splicing switch of AKT2. Principally, RNA splicing is tightly regulated by recruiting RNA-binding proteins (RBPs) that recognize and bind specific regulatory sequences embedded in pre-mRNA transcripts. Hereby, we predicted the potential RBPs interacting with AKT2 by

extracting RBP binding sites from the POSTAR2 database (Zhu et al, 2019). A total of 17 potential RBPs were identified (Dataset EV2). Among these RBPs, hnRNPK, NCBP2, hnRNPC, and hnRNPM showed obvious correlations with the AKT2 splicing isoforms AKT2-210 and AKT2-206 in expression levels (Fig. 6A). To identify the RBPs involved in processing AKT2 splicing, we efficiently knocked down the RBPs in both parental and resistant SK-MEL-28 as well as A375 cells, respectively (Appendix Fig. S13). The results indicated that only knockdown of hnRNPK consistently

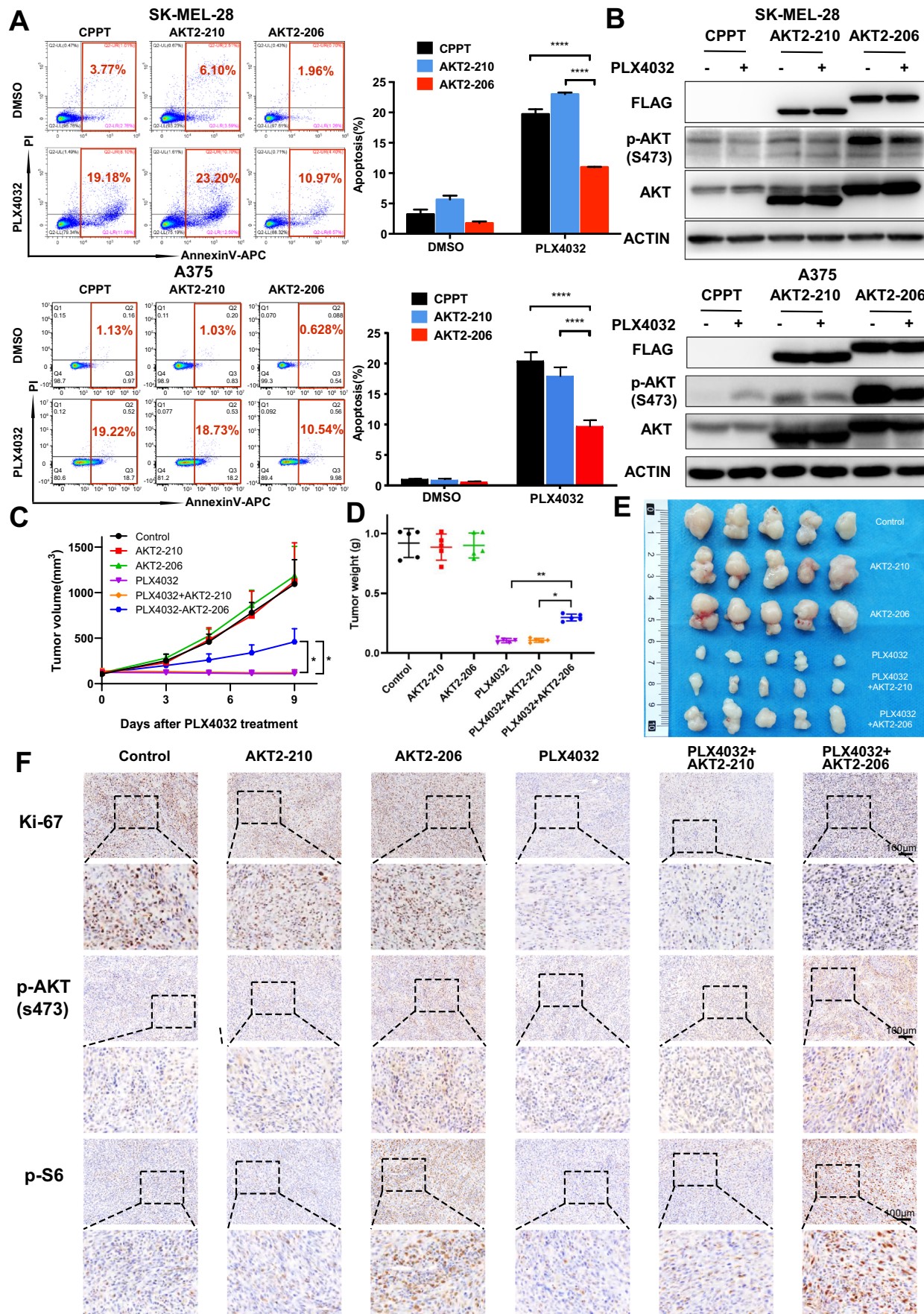

**Figure 5.   AKT2-206 induces BRAFi resistance in melanoma.**

(**A**) Overexpression of AKT2-206 significantly reduced apoptosis induced by PLX4032 (1 μM, 48 h) in SK-MEL-28 and A375 cells. The data are presented as mean ± SD ($n = 3$, biological replicates); ANOVA, SK-MEL-28, AKT2-206 vs. CPPT ($P < 0.0001$), AKT2-206 vs. AKT2-210 ($P < 0.0001$); A375, AKT2-206 vs. CPPT ($P < 0.0001$), AKT2-206 vs. AKT2-210 ($P < 0.0001$). (**B**) The effect of PLX4032 (1 μM, 48 h) on PI3K-AKT signaling in the presence of AKT2-206 or AKT2-210 overexpression was assessed by western blotting. (**C**) The average tumor growth curves for the mouse model xenografted with SK-MEL-28 cells transfected with vector, AKT2-210, and AKT2-206 with or without PLX4032. The quantitative results are presented as the mean ± SD ($n = 5$, biological replicates); ANOVA, PLX4032 vs. PLX4032 + AKT2-206 ($P = 0.027$), PLX4032 + AKT2-210 vs. PLX4032 + AKT2-206 ($P = 0.036$). (**D**) Tumor weights were measured following euthanasia after the final administration of the drug. Quantitative results are presented as the mean ± SD ($n = 5$, biological replicates); ANOVA, PLX4032 vs. PLX4032 + AKT2-206 ($P = 0.0024$), PLX4032 + AKT2-210 vs. PLX4032 + AKT2-206 ($P = 0.011$). (**E**) Representative images of tumors from each group. (**F**) Representative immunohistochemical images of Ki67, p-AKT (S473), and p-S6 staining in tumors from each group. Scale bar = 100 μm. *$P < 0.05$, **$P < 0.01$, ****$P < 0.0001$. Source data are available online for this figure.

attenuated the expression level of AKT2-206 in parental and resistant SK-MEL-28 and A375 cells (Fig. 6B,C). Notably, the reduction in AKT2-206 transcription was accompanied by a decrease in the PSI of exon 10, defined as the percentage of mRNA containing exon 10 relative to the total AKT2 mRNA. This suggests that hnRNPK may act as an RBP to promote the production of AKT2-206 by regulating the alternative splicing of exon 10 (Fig. 6D). We then assessed the cell apoptosis of SK-MEL-28 and A375 cells with hnRNPK knockdown in the presence of PLX4032. The result showed that hnRNPK knockdown augmented the cell apoptosis induced by PLX4032 (Fig. 6E). To elucidate whether hnRNPK mediates BRAFi resistance by modulating the alternative splicing of AKT2, we overexpressed AKT2-206 or AKT2-210 in SK-MEL-28 and A375 cells with hnRNPK knockdown, in the presence of PLX4032. The cell apoptosis assay demonstrated that the additional overexpression of AKT2-206, but not AKT2-210, partially rescued the cell apoptosis induced by hnRNPK knockdown (Fig. 6E; Appendix Fig. S14).

To determine whether A-loop-dependent AKT signaling remodeling is conserved during therapy resistance, we investigated the perturbation of AKT2 alternative splicing by BRAFi treatment in a mouse model. SMM102, a murine melanoma cell line harboring a *BRAF* mutation (Liu XW et al, 2019), was transplanted into both flanks of C57BL/6 mice, followed by continuous PLX4032 treatment. Tumor volumes were measured at the indicated time points representing a specific status of drug resistance evolution (Appendix Fig. S15A). The transcriptomes of tumors from the control group and BRAFi-relapsed group were profiled by next-generation RNA sequencing. All Akt2 isoforms expressed in the tumors were identified, with structural domain annotation revealing that mouse Akt2-210 and Akt2-201, respectively, lacked or retained the A-loop catalytic domain (Appendix Fig. S15B). Strikingly, despite differences in exon usage between species (Appendix Fig. S15C), mouse tumor samples also showed conserved depletion of A-loop-deficient isoforms (Akt2-210) and enrichment of A-loop-intact isoforms (Akt2-201) during resistance (Appendix Fig. S15D). Crucially, this splicing switch converged on activation of AKT signaling characterized by elevated p-AKT and p-S6 in resistant tumors (Fig. 6F,G), mirroring what was observed in MAPKi-resistant melanoma cell lines. Mechanistically, while the precise splicing elements driving the splicing switch may differ between species, the hnRNPK elevation observed in resistant mouse tumors correlated strongly with both Akt imbalance and PI3K-AKT pathway activation (Fig. 6F,G). This functional triad—hnRNPK upregulation, A-loop restoration via splicing, and AKT/S6 phosphorylation—bridges species-specific transcript diversity to conserved signaling adaptations. Bioinformatic analysis using

POSTAR3 (Zhao et al, 2022) further predicted potential Hnrnpk binding motifs in mouse Akt2 splicing regulatory regions (Table EV4). Furthermore, in vitro experiments confirmed that knockdown of Hnrnpk in SMM102 cells significantly reduced Akt2-201 mRNA levels (Appendix Fig. S15E). Importantly, these preclinical findings align with clinical evidence showing HNRNPK elevation in 75% of progressed patient biopsies (3/8 with dual elevation of HNRNPK and p-AKT, Fig. 6H; Table EV5). These parallel observations across species reinforce that hnRNPK-mediated alternative splicing serves as a conserved strategy to reactivate A-loop-dependent AKT signaling during MAPKi resistance, despite species-specific splicing patterns. Collectively, our results demonstrate that the hnRNPK-regulated splicing program represents a novel mechanism driving MAPK pathway inhibitor resistance, providing translational insights for targeting splicing-mediated kinase rewiring in refractory melanoma.

## Discussion

Decades of study have supported the notion that individual splicing isoforms play critical roles in various aspects of tumor biology, such as tumor proliferation, invasion, migration, and metastasis. Despite numerous endeavors to understand the molecular rules of alternative splicing in drug resistance, the underlying regulatory mechanisms remain poorly elucidated. In this study, we aimed to provide a comprehensive understanding of genome-wide alterations in splicing isoforms that confer acquired drug resistance in melanoma. We profiled splicing isoforms from baseline to progressive biopsies of MAPKi-treated melanomas with two independent transcriptome sequencing datasets. Annotated and de novo identified isoforms harboring impaired domain alterations were significantly enriched in the PI3K-AKT and MAPK pathways. Splicing alterations of components encompassed in the canonical and three non-canonical modules in the MAPK pathway were identified. We then functionally validated CD14 as a driver gene that confers drug resistance. Notably, we observed a splicing switch of AKT2 from isoform 210 to 206 in the PI3K-AKT pathway, resulting in the restoration of the A-loop in an evolutionarily conserved activation segment. Furthermore, we correlated AKT2-206 splicing switching with known MAPKi resistance drivers (e.g., *NRAS* mutations, *BRAF* splice variants, *NF1* loss) (Di Leo et al, 2024; Huang et al, 2023; Hugo et al, 2015b; Shao and Aplin, 2010). Results showed no significant association between AKT2-206 upregulation and known genomic, transcriptional, or epigenetic resistance mechanisms ($P > 0.05$, Appendix Fig. S16A,B). Strikingly, 5.00% (2/40) of advanced melanomas exhibited upregulated

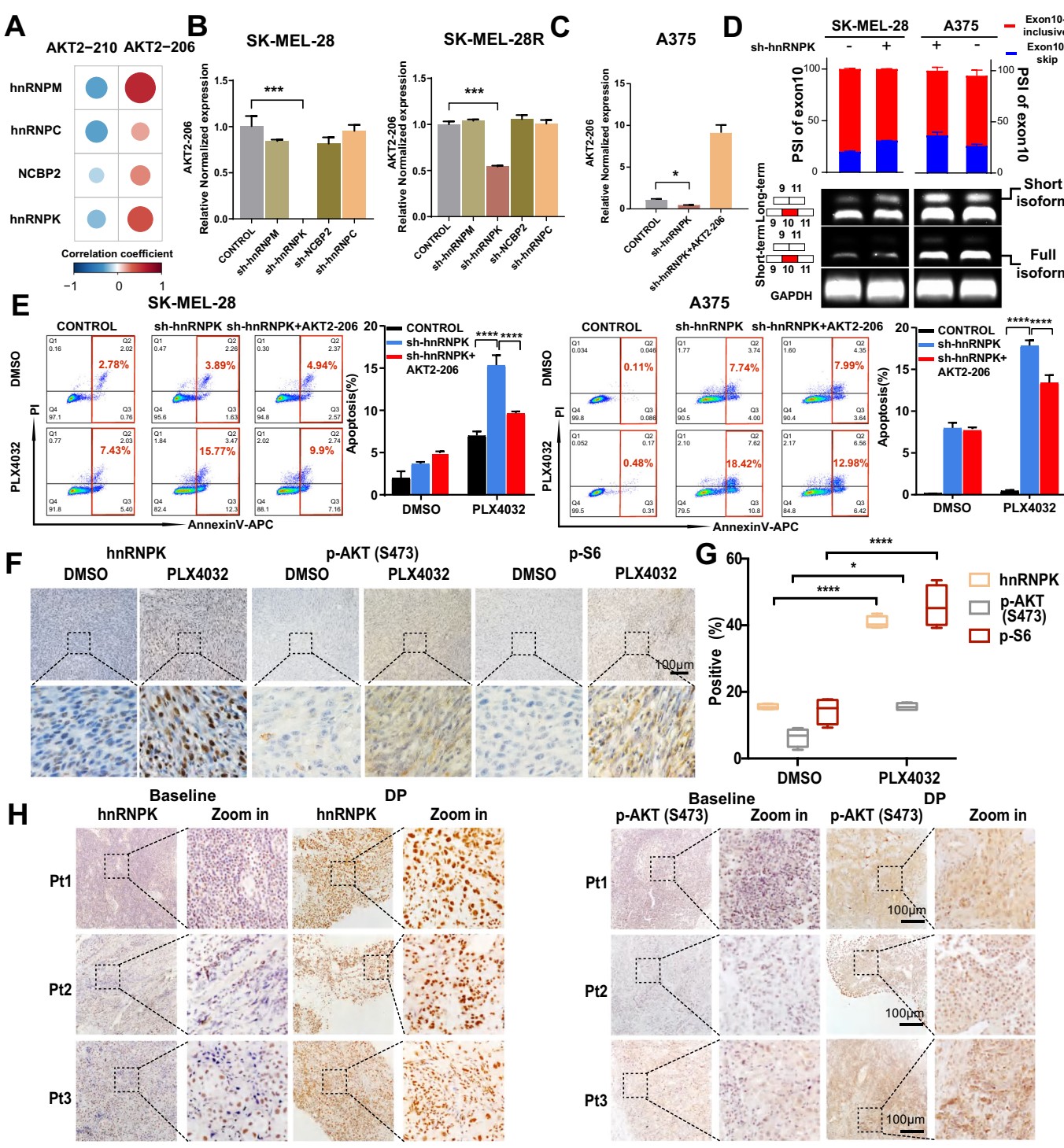

AKT2-206 expression in the absence of established resistance drivers, suggesting that it may serve as an independent resistance mechanism in molecularly distinct patient subgroups. The alternative splicing of AKT2 exon 10 was regulated by a spliceosome complex that contained hnRNPK. Although the binding sites of hnRNPK to AKT2 are distal to introns 9–10 or exon 10 of AKT2 (Appendix Fig. S17), as a multifunctional RNA-binding protein, hnRNPK may still regulate alternative splicing of exon 10 through

mechanisms such as remodeling of RNA secondary/tertiary structures (Jones et al, 2022), recruiting core splicing machinery or competing with repressive factors (Kong et al, 2024; Naganuma et al, 2012), indirectly modulating splicing by affecting the expression or activity of other splicing regulators (Peng et al, 2019), and coupling with transcriptional dynamics (Ip et al, 2011; Mikula et al, 2013). Functionally, ectopic expression of AKT2-206 conferred MAPKi resistance in sensitive cell lines, while

◄ **Figure 6. hnRNPK acts as a key regulator of AKT2 splicing switch.**

(A) The splicing factor hnRNPK binding to AKT2 transcripts positively correlates with AKT2-206. The magnitude of the point represents the correlation coefficient. Positive correlations are displayed in red, and negative correlations in blue. (B) The expression levels of AKT2-206 after knockdown of candidate splicing factors. The data are shown as mean ± SD ($n = 3$, biological replicates); ANOVA, SK-MEL-28 ($P = 0.0008$), SK-MEL-28R ($P = 0.0002$). (C) The expression levels of AKT2-206 after knockdown of hnRNPK and overexpression of AKT2-206 in A375 cells. The data are shown as mean ± SD ($n = 3$, biological replicates); ANOVA, $P = 0.015$. (D) RT-PCR analysis of AKT2 exon 10-inclusive and exon 10-skip isoforms before and after hnRNPK knockdown. Long-term exposure was used to observe changes in PCR products that do not contain exon 10, and short-term exposure was used to observe changes in PCR products that contain exon 10. The data are shown as mean ± SD ($n = 3$, biological replicates). (E) Induction of anti-apoptotic effects upon overexpression of AKT2-206 with or without PLX4032 (1 μM, 96 h). The results were quantified and presented as bar plots. The data are shown as mean ± SD ($n = 3$, biological replicates); ANOVA, SK-MEL28, sh-hnRNPK vs. CONTROL ($P < 0.0001$), sh-hnRNPK+AKT2-206 vs. sh-hnRNPK ($P < 0.0001$); A375, sh-hnRNPK vs. CONTROL ($P < 0.0001$), sh-hnRNPK+AKT2-206 vs. sh-hnRNPK ($P < 0.0001$). (F) Representative images of hnRNPK, p-AKT (S473), and p-S6 by IHC staining in SMM102 tumors with or without MAPKi treatment (scale bar = 100 μm). (G) Box plot showing the quantitative data of IHC staining for hnRNPK, p-AKT (S473), and p-S6. Boxes show median (center line) and 25–75th percentiles (bounds), whiskers indicate 1.5 × interquartile range. The data are shown as mean ± SD ($n = 4$, biological replicates); ANOVA, hnRNPK ($P < 0.0001$), p-AKT(S473) ($P = 0.029$), p-S6 ($P < 0.0001$). (H) Representative images of hnRNPK and p-AKT (S473) by IHC staining in baseline and DP samples from melanoma patients receiving MAPKi therapy, scale bar = 100 μm. *$P < 0.05$, ***$P < 0.001$, ****$P < 0.0001$. Source data are available online for this figure.

knockdown of hnRNPK enhanced the sensitivity of tumor cells to MAPKi. Consistently, the Akt2 splicing switch from Akt2-210 to Akt2-201 in MAPKi-treated mouse model recapitulates the splicing pattern mediating A-loop restoration observed in melanoma patients. Notably, this splicing switch in mice involves different splicing events that are different from those in melanoma patients. Despite species differences in splicing mechanisms, this switch is still significantly associated with hnRNPK upregulation, A-loop recovery, and PI3K-AKT pathway activation. Importantly, pre-clinical findings align with clinical data showing HNRNPK/p-AKT co-elevation in 37.5% of progressing patients ($n = 8$), underscoring the conservation of hnRNPK-mediated A-loop-dependent AKT kinase hyperactivity in MAPKi resistance despite interspecies splicing diversity.

The mechanisms of resistance to MAPKi, including BRAFi and BRAFi+MEKi, had been systematically investigated at the genetic and transcriptomic levels by our group (Hugo et al, 2015b) and Dr. Chin's group (Kwong et al, 2015) several years ago. While these studies defined canonical resistance mechanisms, emerging evidence underscores alternative splicing as a pervasive adaptive strategy enabling malignant escape from therapeutic pressure (Aya et al, 2024; Dewaele et al, 2016). However, prior studies focused on single-gene splicing events. In this study, we provided a comprehensive understanding of the mechanisms by which melanoma cells acquire drug resistance via hijacking genome-wide mRNA splicing. A series of aberrant alternative splicing isoforms with impaired functional domains were identified from patient-matched baseline and disease-progressive biopsies. Notably, the identified isoforms were significantly enriched in two predominant oncogenic pathways, MAPK and PI3K-AKT, which are related to melanoma pathology and MAPKi resistance development. More importantly, this study expanded our understanding of this type of drug resistance mechanism from the individual gene level to pathways and even the entire genome-wide scope.

In the past decade, the resistance mechanisms of MAPK-targeted therapy in melanoma have been extensively investigated. Multiple mechanisms at the levels of epigenetics, genetics, transcription, post-transcriptional modification, translation, etc., have been documented, providing biomarkers for clinical personalized precision medicine in terms of the development of novel combinatorial regimens. Our findings suggest that splicing switch-driven signaling restoration may represent an important approach. Since this splicing switch only slightly or does not alter the total mRNA abundance of individual genes, it cannot be detected at the gene level. Therefore, it is necessary to detect this alteration in clinical diagnosis, especially for patients who have experienced drug relapse.

## Methods

**Reagents and tools table**

| Reagent/resource | Reference or source | Identifier or catalog number |
|---|---|---|
| **Experimental models** | | |
| BALB/c (*M. musculus*) | Beijing Huafukang Biotechnology Co., Ltd. (Beijing, China) | N/A |
| C57BL/6 | Beijing Vital River Laboratory Animal Technology Co., Ltd. | N/A |
| SK-MEL-28 | Core Facility of West China Hospital | Cat #HTB-72 |
| A375 | Core Facility of West China Hospital | Cat #CRL-1619 |
| SMM102 | Core Facility of West China Hospital | *Braf*$^{V600E/wt}$, *Cdkn2a*$^{-/-}$, *Pten*$^{-/-}$ |
| HEK293T | Core Facility of West China Hospital | Cat #CRL-3216 |
| **Recombinant DNA** | | |
| pLKO.1 puro | This paper | N/A |
| pRRL.sin.cPPT.MCS.IRES.GFP | This paper | N/A |
| PMDLGg/p | This paper | N/A |
| RSV-REV | This paper | N/A |
| PMD.G | This paper | N/A |
| **Antibodies** | | |
| Anti-p-S6 | Cell Signaling Technology | Cat #5364 |
| Anti-p-AKT (S473) | Cell Signaling Technology | Cat #9271 |
| Anti-AKT | Cell Signaling Technology | Cat #9272 |
| Anti-FLAG | Sigma | Cat #F1804 |
| Anti-Actin | ZSGB-BIO | Cat #TA-09 |

| Reagent/resource | Reference or source | Identifier or catalog number |
|---|---|---|
| Anti-hnRNPK | Abcam | Cat #ab52600 |
| Anti-S6 | Abcam | Cat #ab60948 |
| Anti-p-AKT | Cell Signaling Technology | Cat #2965L |
| **Oligonucleotides and other sequence-based reagents** | | |
| sh-hnRNPK | This paper | CCAACATTCCTCTGCTTCA |
| sh-hnRNPC | This paper | GGATGACGACGATAAGTAAGG |
| sh-hnRNPM | This paper | GCTGTGCAAGCTATATCTATG |
| sh-NCBP2 | This paper | ACGCCATGCGGTACATAAATG |
| qPCR primer of hnRNPK | This paper | F: GGCTCTCCGTACAGACCATATT R: AGCATTCCACAGCATCAGATTCG |
| qPCR primer of hnRNPC | This paper | F: GCAGAGCCAAAAGTGAACCG R: ACGTTTCGAGGGCACTACAG |
| qPCR primer of hnRNPM | This paper | F: AGGCTGGAAGACTTGGAAGC R: TAGCAGCTGGCCATTGAACA |
| qPCR primer of NCBP2 | This paper | F: TCATTCGCACAGACTGGGAC R: TCTCCCAGCATCGTAGTCCT |
| qPCR primer of AKT2 | This paper | F: AAAGAAGGCTGGCTCCACAA R: TCGCTCTTCAGCAGGAAGT |
| qPCR primer of AKT2-206 | This paper | F: AAAGAGGGCATCAGTGACGG R: CCTGGTTGTAGAAGGGCAGG |
| RT-PCR primer of GAPDH | This paper | F:GATTCCACCCATGGCAAATTC R:CTGGAAGATGGTGATGGGATT |
| RT-PCR primers for the detection of alternative splicing of exon 10 | This paper | F: CTCGGCTCTTGAGTACTTG R: CCATAGTCATTGTCCTCCAG |
| ASO of exon 10 | This paper | GTTTTCATGGTGGCCCCGTC |
| Mus sh-hnRNPK | This paper | CAGTGCTGATATTGAGACGAT |
| qPCR primer of Akt2-210 | This paper | F: ATGCAGAGGTCTTGGAAAAG R: ATACAGTATCGTTGCACACA |
| qPCR primer of Akt2-201 | This paper | F: GCTGAAAGGAGACTGTAAAA R: TACAGTATCGTCTTGGGTCT |
| qPCR primer of Akt2-205 | This paper | F: TAAACAGCTTCTGTGTGGAA R: GCAGCTGTAATCAAATGTCC |
| qPCR primer of Akt2-214 | This paper | F: ATGTCTCTCCCTCCTCAATA R: AGTAAGGAAAGGCCAGCTAT |
| **Chemicals, enzymes, and other reagents** | | |
| PLX4032 | Selleck | Cat #S1267 |
| DMSO | Sigma | Cat #67-68-5 |
| CCT128930 | Selleck | Cat #S2635 |
| Capivasertib | Selleck | Cat #S8019 |
| **Software** | | |
| FastQC | N/A | https://www.bioinformatics.babraham.ac.uk/projects/fastqc/ |
| Trimmomatic | Bolger et al, 2014 | http://www.usadellab.org/cms/?page=trimmomatic |
| STAR | Dobin et al, 2013 | https://github.com/alexdobin/STAR |
| AIDE | Li et al, 2019 | https://github.com/Vivianstats/AIDE |
| StringTie | Pertea et al, 2015 | https://ccb.jhu.edu/software/stringtie/ |
| Cufflinks | Trapnell et al, 2012 | https://cole-trapnell-lab.github.io/cufflinks/ |
| rMTAS | Shen et al, 2014 | https://github.com/Xinglab/rmats-turbo |
| Clustal Omega | Sievers and Higgins, 2018 | https://www.ebi.ac.uk/jdispatcher/msa/clustalo |
| SWISS-MODEL | Waterhouse et al, 2018 | https://swissmodel.expasy.org/ |
| SplicingFactory | Dankó et al, 2022 | https://github.com/esebesty/SplicingFactory |

| Reagent/resource | Reference or source | Identifier or catalog number |
|---|---|---|
| GraphPad Prism | N/A | N/A |
| FlowJo | N/A | N/A |
| Image Lab | N/A | N/A |
| R | Open source | https://cran.r-project.org/ |
| R Studio | Open source | https://posit.co/ |
| **Other** | | |
| Lipofectamine™ RNAiMAX Transfection Reagent | Thermo Fisher Scientific | Cat #13778100 |
| Annexin V Apoptosis Detection Kit | 4A Biotech | Cat #FXP018 |
| APPRIS database | Rodriguez et al, 2018 | https://appris.bioinfo.cnio.es/#/ |
| Ensembl database | Hubbard et al, 2002 | https://useast.ensembl.org/index.html |
| DriverDB database | Liu et al, 2020 | http://ngs.ym.edu.tw/driverdb |
| RNAactDrug database | Dong et al, 2020 | http://bio-bigdata.hrbmu.edu.cn/RNAactDrug |
| OncoKB database | Chakravarty et al, 2017 | https://www.oncokb.org/ |
| GEPIA database | Tang et al, 2019 | http://gepia.cancer-pku.cn/ |
| DIGGER database | Louadi et al, 2021 | https://exbio.wzw.tum.de/digger/ |
| KOBAS database | Xie et al, 2011 | http://kobas.cbi.pku.edu.cn/ |
| POSTAR2 database | Zhu et al, 2019 | http://lulab.life.tsinghua.edu.cn/postar |
| POSTAR3 database | Zhao et al, 2022 | http://111.198.139.65/ |
| DcGO database | Fang and Gough, 2013 | http://supfam.org/SUPERFAMILY/dcGO |
| Melanocytes dataset | Data ref: Hanniford et al, 2020a | GSE138711 |
| Nevus and primary melanoma dataset | Data ref: Badal et al, 2017a | GSE98394 |
| Baseline and MAPKi resistance of melanoma dataset | Data ref: Hugo et al, 2015a | GSE65186 |
| Baseline and MAPKi resistance of melanoma dataset | Data ref: Kwong et al, 2015 | EGAS00001000992 |
| Baseline and MAPKi resistance of melanoma dataset | Data ref: Song et al, 2017b | GSE75299 |
| Baseline and MAPKi resistance of melanoma dataset | Data ref: Kang et al, 2023a | GSE203545 |
| Baseline and MAPKi resistance of melanoma dataset | Data ref: Riordan et al, 2024 | GSE285131 |
| Baseline and MAPKi resistance of melanoma dataset | Data ref: Song et al, 2017c | GSE103630 |
| Baseline and MAPKi resistance of melanoma dataset | Data ref: Patel et al, 2021a | GSE161430 |
| Baseline and MEKi resistance of melanoma dataset | Data ref: Xu et al, 2022a | GSE161430 |

## Ethics

All specimens were obtained with appropriate informed consent from the patients. This study followed the principles of the WMA Declaration of Helsinki was approved by the Institute Research

Ethics Committee of Peking University Cancer Hospital and Institute. All animal studies were performed under the institutional ethics guidelines for animal experiments which were approved by the Experimental Animal Ethics Committee of the West China Hospital, Sichuan University.

## Data preparation

To determine the dynamics of splicing isoforms throughout melanoma oncogenesis and MAPKi resistance evolution in depth, we collected publicly available RNAseq data from NCBI GEO database (GSE138711 (Data ref: Hanniford et al, 2020a; Hanniford et al, 2020b), GSE98394 (Badal et al, 2017b; Data ref: Badal et al, 2017a), GSE65186 (Data ref: Hugo et al, 2015a; Hugo et al, 2015b)), and European Nucleotide Archive database (EGAS00001000992 (Data ref: Kwong et al, 2015; Kwong et al, 2015)), containing data from patients with melanoma who received MAPK inhibitor regimens (BRAFi or BRAFi+MEKi). These specimens consisted of 44 disease-progressive melanomas (DPs), and 21 patient-matched pre-treatment melanomas (Baseline). Independent MAPKi-treated melanoma dataset from the NCBI GEO database, including GSE75299 (Data ref: Song et al, 2017b; Song et al, 2017a), GSE203545 (Data ref: Kang et al, 2023a; Kang et al, 2023b), GSE285131 (Data ref: Riordan et al, 2024), GSE103630 (Data ref: Song et al, 2017b; Song et al, 2017a), GSE186108 (Data ref: Patel et al, 2021a; Patel et al, 2021b). Tumor sequencing data of mouse models are from the NCBI GEO database, GSE161430 (Xu et al, 2022a; Xu et al, 2022b).

## Identification and quantification of splicing isoforms

Raw data were downloaded from public databases and analyzed using a custom-tailored analysis pipeline. Briefly, we used FastQC v0.11.6 for quality control and Trimmomatic v0.36 (Bolger et al, 2014) to remove adapter sequences and trim low-quality bases. Following trimming, reads were aligned to the human reference genome (Gencode v29) using STAR v2.5.0a (Dobin et al, 2013). Only reads that were uniquely mapped to the genome were used for downstream analysis. Samples with low mapping rates were excluded. Ultimately, 65 tumor biopsies (21 Baselines, 44 DPs) from 21 patients were retained for downstream analysis.

Splicing isoforms were reconstructed by using AIDE v1.0.0 (Li et al, 2019), known for its superior performance in isoform discovery and abundance estimation, and quantified by StringTie v1.3.5. The assembled isoforms from each sample were merged into a comprehensive and non-redundant set using StringTie (Pertea et al, 2015) "--merge" mode with the -G option. Finally, we obtained a splicing isoform expression matrix, comprising 96,998 known and 74,532 novel isoforms assembled de novo from the RNAseq data. Transcript isoform diversity was estimated by SplicingFactory v1.6.0 (Dankó et al, 2022).

## Differential expression and functional analysis of splicing isoforms

To identify functional splicing isoforms related to drug resistance, we established a rational approach to screen promising candidates. In detail, we compared the expression profiles of splicing isoforms in DP melanomas with those in patient-matched baseline

biopsies. Only splicing isoforms with ≥10 TPM in at least one sample in the paired comparison group were retained for differential expression analysis. We used |1-FoldChange|>0.2 as the cutoff to define aberrantly expressed splicing isoforms. These splicing isoforms were ranked based on the frequency of abnormal expression in DP samples. Considering the expression concordance, splicing isoforms with high differences of up- and downregulated expression events (the absolute value of the difference >6) were retained.

To characterize the structure and function of splicing isoforms with aberrant expression, we retrieved protein structural and functional features of each splicing isoform from the APPRIS (Rodriguez et al, 2018) and Ensembl (Hubbard et al, 2002) databases. We further explored the potential cancer-related roles of these splicing isoforms through literature mining. Distinct cancer-related gene lists were collected from public databases and previous studies, including cancer driver genes in SKCM from DriverDB (Liu et al, 2020); cancer-, melanoma-, and MAPKi resistance-, and immunotherapy-related genes compiled by Hugo W et al (Hugo et al, 2015b); mRNA molecules exhibiting drug sensitivity to vemurafenib or selumetinib screened in the RNAactDrug database (Dong et al, 2020); and a cancer gene list obtained from the OncoKB database (Chakravarty et al, 2017). The expression profiles of splicing isoforms in the TCGA SKCM cohort were extracted from GEPIA (Tang et al, 2019). Subsequently, we focused on a subset of 25,318 protein-coding splicing isoforms with aberrant expression. The interaction domain information of alternative splicing isoforms was sourced from the DIGGER database (Louadi et al, 2021). To explore the functional consequences of alternative splicing, we extracted sequence variations by comparing protein sequences of the principal isoform and splicing isoform. We then annotated sequence variations based on Pfam domain information and thereby predicted the functional consequences of splicing isoforms. GO annotation and enrichment analysis was performed using DcGO (FDR < 0.05) (Fang and Gough, 2013). Chord diagrams drawn using the R circlize package were used to graphically represent the connections between GO terms and Pfam domains. Pathway enrichment analysis was conducted using KOBAS (FDR < 0.05) (Xie et al, 2011), statistical significance in KOBAS was calculated via over-representation analysis using the hypergeometric test.

## Alternative splicing event quantification

We utilized rMTAS (v4.1.1) algorithm to identify differential alternative splicing (AS) events (Shen et al, 2014). rMTAS can identify five types of AS events: mutually exclusive exons (MXE), alternative 5′ splice site (A5SS), alternative 3′ splice site (A3SS), skipped exon (SE), and retained intron (RI). AS events with |IncLevelDifference|>0.1 and FDR < 0.05 were defined as significantly differentially expressed.

## Homology-based structural analysis

Clustal Omega (Sievers and Higgins, 2018) was used to carry out sequence alignment for AKT2-206 and AKT2-210 proteins. Homology structures of AKT2-206 and AKT2-210 were generated based on the crystal structure of AKT2 (1O6K) (Yang et al, 2002) using SWISS-MODEL (Waterhouse et al, 2018).

## Prediction of candidate RBPs

To identify potential regulators of AKT2 alternative splicing in humans, AKT2-interacting RBPs were then screened through the POSTAR2 database (Zhu et al, 2019). POSTAR2 is a comprehensive database for exploring post-transcriptional regulatory mechanisms coordinated by RBPs, leveraging integrated large-scale high-throughput sequencing datasets and other public resources. As a complementary approach, correlations between AKT2 splicing isoforms and RBPs identified as putative regulators were calculated based on transcriptome profiles.

## Patients and specimens

This study was approved by the Medical Ethics Committee of Peking University Cancer Hospital and adhered to all relevant ethical regulations. Baseline tumor specimens and patient-matched disease-progressive tumors were obtained from melanoma patients undergoing MAPKi therapy. A total of 16 paired samples were collected from 8 melanoma patients to facilitate IHC analysis.

## Cell culture

BRAF-mutant melanoma cell lines SK-MEL-28 and A375 were provided by the Core Facility of West China Hospital, while SMM102 was derived from transgenic $Braf^{V600E/wt}$, $Cdkn2a^{-/-}$, $Pten^{-/-}$ mice, as previously reported (Liu XW et al, 2019). All cell lines were cultured in Dulbecco's modified Eagle's medium (DMEM) supplemented with 10% fetal bovine serum (FBS) at 37 °C under 5% $CO_2$. The PLX4032-resistant $BRAF^{V600E}$ melanoma cell line SK-MEL-28R was generated previously by culturing cells in escalating concentrations of PLX4032. SK-MEL-28R cells were grown in the same media as the parent cell line with the addition of 2 μM PLX4032.

## Generation of knockdown (KD) cell lines

KD cells were generated by infecting melanoma cells with lentiviral particles expressing gene-targeted short hairpin RNAs (shRNAs). Briefly, shRNA sequences were first subcloned into the pLKO.1 puro lentiviral vector and then validated by Sanger DNA sequencing. Modified lentiviral vectors and packaging plasmids (pMDLg/p, pRSV-REV, and pMD.G) were simultaneously transfected into HEK293T cells. Supernatants containing viruses were collected 48 h after transfection, and destination cells were transduced with the viruses for 24 h. Cells stably expressing shRNAs were obtained by puromycin selection.

## Generation of cell lines overexpressing splicing isoforms

Stably expressing cells were generated by infecting SK-MEL-28 and A375 cells with lentiviral particles expressing splicing isoforms. Briefly, the sequences of splicing isoforms were synthesized and cloned into pRRL.sin.cPPT.MCS.IRES.GFP vector. Packaging and infection of viruses were performed as described above.

## RNA isolation and PCR analysis

Total RNA was isolated using TRIzol reagent (Invitrogen). cDNA was prepared using HiScript III RT SuperMix for quantitative real-time PCR (qRT-PCR) following the manufacturer's instructions. qRT-PCR was performed on the CFX96 Real-Time PCR Machine (Bio-Rad). Reactions were performed in 20 μl volumes containing 1× SYBR Green Supermix and 0.4 μM each of the forward and reverse primers. In qPCR thermal cycling, cDNA undergoes denaturation at 95 °C, primer annealing at 60 °C, and extension at 72 °C by DNA polymerase. Fluorescence is measured during each extension phase as the dye incorporates into newly synthesized DNA. This cyclic process typically repeats for 40 cycles, with real-time fluorescence data collection for quantification. PCR products were identified by 2% agarose gel electrophoresis and ethidium bromide visualization.

## Protein detection

Cells were seeded into the six-well plate and treated with PLX4032 at the indicated time points. Then the cells were lysed in RIPA buffer supplemented with protease and phosphatase inhibitor cocktails (Cocktails). The cell lysates were then subjected to SDS-PAGE and transferred to polyvinylidene difluoride (PVDF) membranes (Bio-Rad). Following this, the membranes were blocked with 5% bovine serum albumin (BSA). Each blot was then incubated with the aforementioned primary antibodies. Immunohistochemistry analysis was used to detect the expression of hnRNPK, p-AKT (S473), and p-S6 in both mice tumors and patient tissues. Formalin-fixed and paraffin-embedded (FFPE) tissue sections were rehydrated and antigen-retrieved. Then, the sections were incubated with primary antibodies followed by HRP-conjugated secondary antibodies.

## Cell proliferation assay

The effects of stable overexpression of the indicated splicing isoforms on cellular sensitivity to PLX4032-mediated growth suppression were determined by cell viability assays (3 days of drug exposure) and clonogenic assays (10–14 days). In the cell viability assay, the cells were set up in 96-well plates (1000–2000 cells/well, 5 replicates) for 24 h to adhere and then titrated with varying concentrations of PLX4032 for 72 h. The relative cell numbers were determined by staining the cells with MTS dye and measuring the absorbance at 490 nm. In the clonogenic assay, cells were seeded into six-well plates at a single-cell density, allowed to adhere overnight, and then cultured in the absence or presence of PLX4032 as indicated. After 10–14 days, the cells were fixed with 4% paraformaldehyde and stained with 0.05% crystal violet. Images of the plates were obtained for quantitation.

## Cell apoptosis assay

Cells were seeded at $10^5$ cells/well into six-well plates overnight. Then, cells were treated with or without PLX4032 for 96 h. After that, the cells were collected and stained with Annexin V Apoptosis Detection Kit (4A Biotech) and analyzed by CytoFLEX from Beckman Coulter.

## Animal experiments

All experiments and procedures involving mice were performed in accordance with the guidelines of the Animal Care Committee of Sichuan University. Six- to eight-week-old BALB/c nude mice were obtained from Beijing Huafukang Biotechnology Co., Ltd. (Beijing, China). The 100 μl mixture of $5 × 10^6$ SK-MEL-28 cells transfected with vector, AKT2-210, and AKT2-206 was subcutaneously injected into the right-side dorsal

flank of the mouse. Tumor volumes were measured with the caliper every two days and calculated using the following formula: tumor volume $(mm^3) = (Length \times Width^2)/2$. When the average tumor size reached ~100 mm³, the mice inoculated with the same cells were randomly divided into two groups with five mice in each group. Then, the mice were treated with vehicle or PLX4032 (50 mg/kg, dissolved in 5% DMSO, 40% PEG300, and 5% Tween-80 in distilled water, 5 days per week) by intraperitoneal injection. Tumors were weighed, collected, and fixed in formalin for immunohistochemistry (IHC).

Six-week-old female C57BL/6 mice (Beijing Vital River Laboratory Animal Technology Co., Ltd.) were subcutaneously injected with $1 \times 10^5$ SMM102 cells, with two injections per mouse. Mice with tumors measuring 150–300 mm³ were randomly assigned to receive either vehicle (control) or PLX4032 (50 mg/kg, 5 days per week). Tumor dimensions were measured with a caliper every 3 days. Tumor volumes were calculated using the following formula: $(Length \times Width^2)/2$. The transcriptomes of tumors from the control group and BRAFi-relapsed group (day 27) were profiled using RNAseq. The raw sequencing data were subjected to quality control and removal of low-quality bases, as previously described. For the clean sequencing reads, Kallisto v0.46.2 (Bray et al, 2016) was employed for the quantification of alternative splicing isoforms. In addition, tumors were fixed in formalin for IHC.

## Statistical analysis

Statistical analysis was conducted using GraphPad Prism 9.0 software and the R package. Replicate data were presented as mean ± SD. To assess differences among groups, an unpaired two-tailed $t$ test or one-way ANOVA was utilized. Statistical significance was determined at a $P$ value of less than 0.05.

# Data availability

No primary datasets have been generated and deposited.

The source data of this paper are collected in the following database record: biostudies:S-SCDT-10_1038-S44319-025-00521-6.

# Peer review information

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

## Acknowledgements

The authors thank members of the Shi laboratory for helpful discussions. This work was supported by (1) the National Key Research and Development Program of China (2022YFA1207300 [2022YFA1207303]); (2) National Natural Science Foundation of China (No. 82404682, No. 82172634, No. 81902792, No. 82304527, No. 82102492); (3) Key Program of the Science and Technology Bureau of Sichuan (No. 2021YFSY0007); (4) 1.3.5 project for disciplines of excellence, West China Hospital, Sichuan University (No. ZYGD23028); (5) the China Post-doctoral Science Foundation (No. 2024M752260, No. 2022M722129, No. 2023M742482); (6) the China Postdoctoral Science Foundation under Grant Number GZC20241148; (7) Sichuan Science and Technology Program (No. 2024NSFSC1725, No. 2024NSFSC1765); (8) Post-doctoral Research Project, West China Hospital, Sichuan University (No. 2024HXBH138).

## Author contributions

**Jing Yu**: Conceptualization; Data curation; Formal analysis; Investigation; Visualization; Methodology; Writing—original draft; Writing—review and editing. **Xiujing He**: Conceptualization; Formal analysis; Investigation; Visualization; Methodology; Writing—original draft; Writing—review and editing. **Xueyan Wang**: Validation; Investigation; Visualization; Writing— original draft; Writing—review and editing. **Chune Yu**: Investigation; Visualization; Writing—original draft. **Xian Jiang**: Investigation; Visualization. **Yanna Li**: Investigation. **Xinyu Liu**: Investigation; Visualization. **Ya Luo**: Resources. **Xuemei Chen**: Resources. **Sisi Wu**: Resources. **Lu Si**: Resources; Supervision. **Jing Jing**: Resources; Supervision. **Xuelei Ma**: Resources; Supervision. **Hubing Shi**: Conceptualization; Resources; Data curation; Formal analysis; Supervision; Funding acquisition; Methodology; Writing—original draft; Project administration; Writing—review and editing.

Source data underlying figure panels in this paper may have individual authorship assigned. Where available, figure panel/source data authorship is listed in the following database record: biostudies:S-SCDT-10_1038-S44319-025-00521-6.

## Disclosure and competing interests statement

The authors declare no competing interests.

