## [Peer Review File · EMBO Reports]

Transcriptome-wide decoding the roles of aberrant splicing in melanoma MAPK-targeted resistance evolution

Hubing Shi, Jing Yu, Xiuqing He, Xueyan Wang, Chune Yu, Xian Jiang, Yanna Li, Xinyu Liu, Ya Luo, Xuemei Chen, Sisi Wu, Lu Si, Jing Jing, and Xuelei Ma

Corresponding author(s): Hubing Shi (shihb@scu.edu.cn), Lu Si (yj3312@wchscu.edu.cn), Xuelei Ma (maxuelei0726@wchscu.cn), Jing Jing (jingjing@wchscu.edu.cn)

Review Timeline:

Submission Date:	26th Dec 24
Editorial Decision:	24th Feb 25
Revision Received:	16th May 25
Editorial Decision:	3rd Jun 25
Revision Received:	5th Jun 25
Accepted:	16th Jun 25

Editor: Esther Schnapp

Transaction Report:

Dear Prof. Shi

Thank you for the submission of your manuscript to EMBO reports. We have now received the full set of referee reports that is pasted below.

As you will see, the referees acknowledge that the findings are interesting. However, they also have several suggestions for how the study could be strengthened. I think all suggestions are good and should be addressed, but please let me know in case you disagree, and we can discuss the exact revision requirements further, also in a video chat, if you like.

I would thus like to invite you to revise your manuscript with the understanding that the referee concerns must be fully addressed and their suggestions taken on board. Please address all referee concerns in a complete point-by-point response. Acceptance of the manuscript will depend on a positive outcome of a second round of review. It is EMBO reports policy to allow a single round of major revision only and acceptance or rejection of the manuscript will therefore depend on the completeness of your responses included in the next, final version of the manuscript.

We realize that it is difficult to revise to a specific deadline. In the interest of protecting the conceptual advance provided by the work, we recommend a revision within 3 months (27th May 2025). Please discuss the revision progress ahead of this time with the editor if you require more time to complete the revisions.

- 1) A data availability section providing access to data deposited in public databases is missing. If you have not deposited any data, please add a sentence to the data availability section that explains that.
- 2) Your manuscript contains statistics and error bars based on $n=2$. Please use scatter blots in these cases. No statistics should be calculated if $n=2$.

5) a complete author checklist, which you can download from our author guidelines <https://www.embopress.org/page/journal/14693178/authorguide>. Please insert information in the checklist that is also reflected in the manuscript. The completed author checklist will also be part of the RPF.

6) Please note that all corresponding authors are required to supply an ORCID ID for their name upon submission of a revised manuscript (<https://orcid.org/>). Please find instructions on how to link your ORCID ID to your account in our manuscript tracking system in our Author guidelines <https://www.embopress.org/page/journal/14693178/authorguide#authorshipguidelines>

7) Before submitting your revision, primary datasets produced in this study need to be deposited in an appropriate public

database (see <https://www.embopress.org/page/journal/14693178/authorguide#datadeposition>). Please remember to provide a reviewer password if the datasets are not yet public. The accession numbers and database should be listed in a formal "Data Availability" section placed after Materials & Method (see also <https://www.embopress.org/page/journal/14693178/authorguide#datadeposition>). Please note that the Data Availability Section is restricted to new primary data that are part of this study. * Note - All links should resolve to a page where the data can be accessed. *

10) Regarding data quantification (see Figure Legends:

<https://www.embopress.org/page/journal/14693178/authorguide#figureformat>)

12) All Materials and Methods need to be described in the main text using our 'Structured Methods' format, which is required for all research articles. According to this format, the Methods section includes a separate Reagents and Tools Table file (listing key reagents, experimental models, software and relevant equipment and including their sources and relevant identifiers) and a Methods and Protocols section describing the methods using a step-by-step protocol format. The aim is to facilitate adoption of the methodologies across labs. More information on how to adhere to this format as well as a downloadable template (.docx) for the Reagents and Tools Table can be found in our author guidelines:

An example of a Method paper with Structured Methods can be found here: <https://www.embopress.org/doi/full/10.1038/s44320-024-00037-6#sec-4>

You are able to opt out of this by letting the editorial office know (emboreports@embo.org). If you do opt out, the Review

Process File link will point to the following statement: "No Review Process File is available with this article, as the authors have chosen not to make the review process public in this case."

I look forward to seeing a revised form of your manuscript when it is ready.

Yours sincerely,

Referee #1:

In the manuscript titled "Transcriptome-wide decoding the roles of aberrant splicing in melanoma MAPK-targeted resistance evolution", the authors developed an analytical approach to identify resistance-specific splicing isoforms using RNA-seq data from melanoma biopsy samples obtained before treatment and during disease progression. As a result, significant splicing changes were observed in genes associated with the MAPK and PI3K-AKT pathways. In particular, a splicing switch in AKT2 was detected in ~29% of disease progression samples. This change was confirmed to cause hyperactivation of AKT2 kinase, leading to resistance to BRAF inhibitors. Overall, this study is well-conducted and presents novel insights into splicing regulation in melanoma resistance. However, my major concern is the extent to which the observed AKT2 splicing switch contributes to resistance. Further clarification on the functional impact of this switch would strengthen the authors' conclusions.

My specific comments are as follows:

Major comments:

1. The potential contribution of the AKT2 splicing switch to resistance should be described in greater detail. It remains unclear from Figures 3D and 4A what type of splicing switch is occurring. Are these mutually exclusive exons, or is another type of alternative splicing involved? In Figure 3E, the splicing pattern change is difficult to interpret. Statistical values supporting the observed switch should be provided to validate its significance.
2. The method used for splicing analysis in this study is relatively uncommon. To strengthen the results, the authors should provide:
 - Delta-PSI (Percent-Spliced-In) values for the AKT2 splicing events.
 - An assessment of the robustness of splicing changes across all significant events.
3. A critical follow-up question is whether this AKT2 splicing event can be therapeutically targeted to enhance sensitivity to MAPK inhibitors. This could be experimentally tested using an antisense oligonucleotide (ASO) specifically targeting the AKT2 splicing event. Such experiments would also help quantify the direct contribution of this splicing switch to resistance.

Addressing the above concerns-particularly by clarifying the functional impact of AKT2 splicing, improving the presentation of statistical data, and considering therapeutic targeting-would significantly enhance the manuscript.

Referee #2:

In this interesting study, the authors investigated the role of alternative pre-mRNA splicing in drug resistance of melanoma to MAPK-targeted therapy. The authors first used RNA-seq analyses to identify a large scale of splicing alterations in MAPKi resistant melanoma and found that the alternatively splicing isoforms are enriched in the MAPK and PI3K-AKT pathways. Then the authors identified a functionally significant splicing isoform in a key member of the AKT family, AKT2. Alternative splicing of exon 10 of AKT2 leads to a switch from isoform AKT2-210 to AKT2-206 and introduces the activated loop (A-loop) to the functional protein, which activates the PI3K-AKT pathway and thereby confers MAPKi resistance. The authors further showed that the alternative splicing of AKT2 exon 10 is potentially directly regulated by RNA splicing factor HNRNPK.

The manuscript is well written. The experiments were well executed and the data clearly presented. This study is novel and reshapes our understanding of the mechanism of drug resistance in melanoma from gene expression changes to alternative splicing (without changes in overall expression levels). Therefore, this study is of major significance in that respect. I think the manuscript will be improved if the authors can address the following comments.

Specific comments:

1. Figures 6B and 6C, the authors showed that the expression of isoform AKT2-206 was reduced upon HNRNPK knockdown. However, it was not clear whether the downregulation of AKT2-206 was caused by alternative splicing of exon 10 of AKT2 or by reduction of the overall transcription of AKT2. The authors may want to perform an RT-PCR analysis to confirm that AKT2 exon 10 is indeed alternatively spliced upon HNRNPK knockdown. Specifically, the authors may perform an RT-PCR reaction (with a

forward primer annealing to the complementary strand of exon 9 and a reverse primer annealing to exon 11) and obtain two PCR products of different sizes in the same reaction (a larger-sized PCR product with exon 10 included and a shorter-sized PCR product without exon 10). Then the authors can estimate the PSI (percent spliced in) of exon 10 by quantifying these two PCR products resolved in an agarose gel. The PSI value of exon 10 is expected to decrease upon HNRNPK knockdown if AKT2 exon 10 is indeed alternatively spliced.

2. The authors predicted seven HNRNPK binding sites in AKT2 transcripts (as shown in Table S5). Are any of these HNRNPK binding sites located in exon 10, intron 9 or intron 10 of AKT2? It would be helpful if the authors could provide a simple illustration of the locations of HNRNPK binding sites relative to the position of exon 10 of AKT2. If the HNRNPK binding sites are not located very close to the splice sites of exon 10, then the authors need to provide an explanation for how the alternative splicing of AKT2 exon 10 is directly regulated by HNRNPK.

3. Alternative splicing is species-specific. The majority of alternative splicing events is not conserved between human and mouse. The authors did not provide any evidence that the mechanism of alternative splicing of human AKT2 exon 10 is conserved in mouse Akt2 exon 10. In fact, according to Figures S11B and S11C, the switch between isoforms Akt2-210 and Akt2-201 is not caused by alternative splicing of exon 10 of mouse Akt2. The authors need to identify the new alternative splicing pattern that leads to the switch between isoforms Akt2-210 and Akt2-201 and then show experimental evidence (e.g., RT-PCR results). Are there HNRNPK binding site(s) in mouse Akt2 transcripts that are potentially responsible for this new alternative splicing event?

4. Figure S11C, it would be helpful if the authors could provide the sequences of the qPCR primers used for detection of the expression of mouse Akt2 isoforms Akt2-210, Akt2-201, Akt2-205 and Akt2-214.

5. Figure 1A, the pie chart of (Melanocyte vs. Nevus) is 100% identical to that of (Nevus vs. Melanoma). The authors need to provide an explanation for this.

6. It would be helpful if the authors could specifically mention in the first paragraph of the Results section that the data shown in Figures 1A and 1B were obtained by using the alternative splicing detection tool, rMATS.

7. Figure 1B, it would be helpful if the authors could clarify A3SS-skip vs. A3SS-inclusive, A5SS-Skip vs. A5SS-inclusive, and MXE-Skip vs. MXE-inclusive. Specifically, does A3SS-Skip mean skipping of the proximal 3' splice site or skipping of the distal 3' splice site? Does A5SS-Skip mean skipping of the proximal 5' splice site or skipping of the distal 5' splice site? Does MXE-Skip mean skipping of the upstream exon or skipping of downstream exon?

Referee #3:

Manuscript: EMBOR-2024-61082V1

Title: "Transcriptome-wide decoding the roles of aberrant splicing in melanoma MAPK-targeted resistance evolution"

The authors assess comprehensively the magnitude of alternative splicing events during melanoma development/progression and MAPKi treatment (targeted therapy) by re-analyzing already published RNA expression data of melanocytes, nevi, and melanomas (baseline vs. targeted therapy). They apply a computational decision tree, allowing the identification of functional splicing isoforms through pathway enrichment analysis while discarding principal isoforms and focusing on repeatedly up/downregulated ones in MAPKi resistant melanoma samples. As probably expected, the MAPK pathway gene set seems to be the most perturbed at the splicing level, but so appears the PI3K-AKT pathway. The authors show evidence for a splicing switch from AKT2 isoform 210 to AKT2 isoform 206 which results in kinase hyperactivity (increase pAKT (S473)) through restoration of the activated fragment A-loop. Functional validation via ectopic overexpression of AKT2 206 and 210 in melanoma cell lines and Xenograft models underlines the role of the isoform AKT2 206 in conferring resistance to BRAF inhibition. Finally, the authors present evidence that the splicing switch from AKT 210 to 206 is mediated by hnRNPk as shRNA-mediated KD of hnRNPk attenuated AKT2-206 expression in parental and resistant SK-MEL-28 and A375 melanoma cells and resensitized them to BRAFi. Lastly, IHC evidence is presented in murine (+/-BRAFi) and human melanomas (baseline vs. DP) showing a co-expression pattern of hnRNPk, pAKT and pS6.

In summary, the manuscript provides solid evidence that the novel splicing switch from AKT2-210 to AKT2-206 confers resistance to BRAFi.

The following points should be addressed to further strengthen the manuscript:

Figure 1A) Piecharts of Melanocyte vs. Nevus and Nevus vs. Melanocytes are redundant, right?

Figure 2A) The middle section of the alluvial plots reads: "Domain retention"? Do you mean "domain retention"?

Figure 2C) The PI3K-AKT pathway has the highest enrichment count (number of DEIs) but a relatively poor p-value, could you comment on that?

Figure 3A,B) What depicts the X axis? Patient samples, if yes, what cohort? doi: 10.1016/j.cell.2015.07.061? Also 3E is this from the same patient cohort?

If yes, could you check if AKT2-206 overexpression is mutually exclusive with genetically driven MAPKi resistance mechanisms? Regarding the frequency of AKT2-206 expression, could you mine alternative melanoma data sets under MAPKi? Or how about

mining the CCLE database as it holds RNA and drug response data? Would you expect to find the splicing switch also in NRAS mutant melanoma challenged with MEKi?

General comment/question: Are both AKT2 isoforms (-210 and -206) equally sensitive to pan-AKTi or AKT2 inhibitors?

Page 10, Line178: "In addition, 11 splicing isoforms corresponding to 10 genes also presented aberrant expression in SKCM cohort from TCGA, suggesting their fundamental roles in tumor progression and drug resistance." I am not sure how far TCGA_SKCM AS-analysis can suggest this as there aren't any drug-treated samples included to my knowledge. What could be compared in TCGA_SKCM is OS, primary vs. metastatic melanoma...

Discussion:

When discussing the role of aberrant splicing in melanoma as means to confer resistance to MAPKi, the authors should include the paper doi.org/10.1172/JCI82534, where authors could show in multiple human melanoma cell lines and PDX mouse models, ASO-mediated skipping of exon 6 decreased MDM4 abundance, inhibited melanoma growth, and enhanced sensitivity to MAPKi.

Dear reviewers,

Thank you for reviewing our manuscript entitled "*Transcriptome-wide decoding the roles of aberrant splicing in melanoma MAPK-targeted resistance evolution*". We would also like to convey our gratitude to the reviewers for their diligent review of our manuscript and providing comments and suggestions to improve the quality of our paper. We have carefully considered each comment and provided our point-by-point response below.

Response to Referee #1:

In the manuscript titled "Transcriptome-wide decoding the roles of aberrant splicing in melanoma MAPK-targeted resistance evolution", the authors developed an analytical approach to identify resistance-specific splicing isoforms using RNA-seq data from melanoma biopsy samples obtained before treatment and during disease progression. As a result, significant splicing changes were observed in genes associated with the MAPK and PI3K-AKT pathways. In particular, a splicing switch in AKT2 was detected in ~29% of disease progression samples. This change was confirmed to cause hyperactivation of AKT2 kinase, leading to resistance to BRAF inhibitors. Overall, this study is well-conducted and presents novel insights into splicing regulation in melanoma resistance. However, my major concern is the extent to which the observed AKT2 splicing switch contributes to resistance. Further clarification on the functional impact of this switch would strengthen the authors' conclusions.

Response: We extend our sincere gratitude to the reviewer for the meticulous evaluation of our work, and are particularly encouraged by the recognition that "his study is well-conducted and presents novel insights into splicing regulation in melanoma resistance." We have systematically addressed each concern through comprehensive supplementary analyses as detailed in the point-to-point responses below. All corresponding revisions have been carefully incorporated into the revised manuscript.

My specific comments are as follows:

Major comments:

1. The potential contribution of the AKT2 splicing switch to resistance should be described in greater detail. It remains unclear from Figures 3D and 4A what type of splicing switch is occurring. Are these mutually exclusive exons, or is another type of alternative splicing involved? In Figure 3E, the splicing pattern change is difficult to interpret. Statistical values supporting the observed switch should be provided to validate its significance.

Response: We sincerely appreciate the reviewer's insightful query regarding the potential contribution of the AKT2 splicing switching in MAPKi resistance. Our expanded analysis and functional validations are summarized below:

(1) AKT2 splicing drives MAPKi resistance via PI3K-AKT signaling

Our functional studies demonstrate that the AKT2 splicing switch directly drives BRAFi resistance by reactivating PI3K-AKT signaling. *In vitro*, overexpression of AKT2-206 (but not AKT2-210) elevated total and phosphorylated AKT levels, suppressed both early and late apoptosis under PLX4032 treatment, and enhanced melanoma cell survival. *In vivo*, AKT2-206 overexpression in xenograft models reversed PLX4032-induced tumor growth inhibition, which is mechanistically attributed to the activation of the PI3K-AKT pathway. In contrast, AKT2-210 exhibited no such pro-survival or signaling effects. As shown in **Response Fig. 2**, exon 10 retention in AKT2-206 restores the catalytically active A-loop conformation, which is absent in AKT2-210. This structural alteration significantly enhances AKT kinase activity. The combined effects of elevated AKT2-206 expression, hyperactivated AKT signaling, and downstream S6 phosphorylation collectively establish a critical signal transduction driving resistance evolution. These data collectively suggest that AKT2-206 can activate the PI3K-AKT pathway, thereby bypassing BRAFi-induced pathway inhibition and driving drug resistance in melanoma cells.

(2) Independence from known resistance mechanisms

To investigate whether the AKT2-206 splicing switch is associated with established MAPKi resistance drivers (Di Leo *et al*, 2024; Huang *et al*, 2023; Hugo *et al*, 2015; Shao & Aplin, 2010) (e.g., NRAS mutations, BRAF splice variants, NF1 loss, **Response Fig. 1A**), we conducted Cramer's V correlation and Fisher's exact test analyses. These analyses revealed no significant association ($p > 0.05$) between AKT2-206 upregulation and known genomic, transcriptional, or epigenetic resistance mechanisms (**Response Fig. 1B**). Furthermore, correlation analyses of *BRAF*^{V600E} alternative splicing with these established resistance drivers also showed no statistical linkage ($p > 0.05$, **Response Fig. 1B**), while the upregulation of PIK3CA mRNA levels was associated with multiple mechanisms, such as the upregulation of mRNA expression levels of genes such as KRAS, NRAS, and AKT3 (**Response Fig. 1B**). Intriguingly, 5.00% (2/40) of progressive melanomas exhibited the upregulated AKT2-206 expression in the absence of established resistance drivers, suggesting its potential as a standalone resistance mechanism in a molecularly distinct patient subset. These findings support the hypothesis that AKT2 splicing conversion represents a distinct resistance pathway.

(3) Therapeutic implications

To directly target this mechanism, we designed an exon 10-specific antisense oligonucleotide (ASO). In SK-MEL-28 and A375 cell lines, ASO-mediated exon 10 silencing significantly sensitized melanoma cells to PLX4032 (**Response Fig. 1C, D**). This confirms the therapeutic potential of targeting the AKT2 exon 10 to overcome MAPKi resistance, with ASO-based exon 10 modulation serving as an actionable strategy to enhance treatment efficacy.

Collectively, our findings underscore the AKT2 splicing switch as a driver of MAPKi resistance and demonstrate ASO-based splice correction as a novel therapeutic approach for melanomas.

Figure for referees not shown.

Response Figure 1. Independence of AKT2 splicing switch from known resistance mechanisms. (A) Recurrence of alterations and heterogeneity mechanistic of known resistance genes and AKT2-206 in acquired resistant tumors. (B) Correlation analysis of AKT2-206, *BRAF*^{V600E} alternative splicing, PIK3CA, and identified drug resistance drivers. Fisher's exact test and Cramer's V correlation. (C) Efficiency of ASO-mediated exon 10 silencing. The data are shown as mean±SD (n=3, technical replicates); ANOVA. (D) Colony formation assay was performed on melanoma cells transfected with negative ASO control and exon 10 ASO with or without PLX4032 (0.01μM). * p < 0.05, **** p < 0.0001.

To clarify the nature of the splicing switch, we have revised Figure 4A to include a

schematic exon structure of AKT2-206 and AKT2-210 (**Response Fig. 2**). This figure now explicitly demonstrates that the two isoforms differ through exon 10 skipping (AKT2-210 skips exon 10, while AKT2-206 retains exon 10). For Figure 3E, we have added statistical validation (paired t-test) to quantify the observed splicing pattern changes (**Response Fig. 3**). The updated analysis confirms that AKT2-206 is significantly upregulated in MAPKi-resistant samples, while AKT2-210 is downregulated, solidifying the robustness of the splicing switch. These revisions collectively validate the exon-skipping-driven splicing switch and reinforce its statistical significance in the context of acquired resistance. We apologize for the initial lack of clarity and have revised both figures and legends to emphasize these critical details.

Figure for referees not shown.

Response Figure 2. Exon composition of AKT2-206 and AKT2-210. Exon 10 skipped in AKT2-210 is highlighted in red. The skipping of exon 10 in AKT2 results in the absence of the activated loop (A-loop). The A-loop is highlighted in gray block, and the regulatory residue Thr 309 is marked with asterisks.

Figure for referees not shown.

Response Figure 3. Expression patterns of AKT2-206 and AKT2-210 in patient-matched pairs. Upregulation and downregulation patterns are displayed in red and blue, respectively. Paired t-test. * $p < 0.05$, ** $p < 0.01$.

Accordingly, we have made additional descriptions on pages 19-20, lines 378-386:

“Furthermore, we correlated AKT2-206 splicing switching with known MAPKi resistance drivers (e.g., NRAS mutations, BRAF splice variants, NF1 loss) (Di Leo et al, 2024; Huang et al, 2023; Hugo et al., 2015; Shao & Aplin, 2010). Results showed no significant association between AKT2-206 upregulation and known genomic, transcriptional, or epigenetic resistance mechanisms ($p > 0.05$, Appendix Fig. S16A, B). Strikingly, 5.00% (2/40) of advanced melanomas exhibited upregulated AKT2-206 expression in the absence of established resistance drivers, suggesting that it may serve as an independent resistance mechanism in molecularly distinct patient subgroups.”

Correspondingly, we have integrated this result into Fig. 3E, 4A and Appendix Fig. S16A, B.

2. *The method used for splicing analysis in this study is relatively uncommon. To strengthen the results, the authors should provide:*

-Delta-PSI (Percent-Spliced-In) values for the AKT2 splicing events.

-An assessment of the robustness of splicing changes across all significant events.

Response: Thank you for your constructive feedback. In response to your comment, we calculated the log2FC of the AKT2-206 isoform and the Delta-PSI for the exon 10

in AKT2. A significant correlation was observed between the two ($R = 0.41$, p value = 0.038, **Response Fig. 4A**), suggesting that changes in AKT2-206 expression are closely linked to exon 10 splicing regulation.

We sincerely appreciate the reviewer's critical comment regarding the robustness of the splicing analysis. To address this concern, we adopted a dual-layered validation strategy integrating technical consistency checks and biological relevance assessments. First, to evaluate the global consistency of splicing changes, we quantified transcript isoform diversity using SplicingFactory-derived Laplace entropy scores (ranging from 0 [single isoform dominance] to 1 [balanced isoform expression]) across all tumor samples. Genes with differential splicing events (identified by rMATS) exhibited markedly higher Laplace entropy scores than those without splicing changes ($p < 1e-5$, **Response Fig. 4B, C**), demonstrating that alternative splicing correlates with increased isoform heterogeneity and validating the technical robustness of the splicing changes we identified. Second, to interrogate the biological significance of these splicing changes, we performed KEGG pathway enrichment analysis (via KOBAS) on genes harboring differential splicing events. These genes were strongly enriched in signal transduction pathways associated with MAPKi resistance, including mTOR and PI3K-AKT signaling pathways ($p < 0.05$, **Response Fig. 4D**). Crucially, this functional link was further corroborated by independent validation. We rigorously screened public repositories (e.g., GEO, ENA) and compiled transcriptomic sequencing data from five independent studies (GSE75299, GSE203545, GSE285131, GSE103630, GSE186108), encompassing 26 matched baseline and post-MAPKi-resistant samples. Strikingly, AKT2-206 expression was elevated ($FC > 1.2$) in 57.7% (15/26) of acquired resistance samples, strongly supporting the robustness of the splicing analysis method we used (**Response Fig. 4E**). This multi-dimensional validation strategy, integrating entropy-based technical validation, pathway-driven functional analysis and cross-study reproducibility checks, confirms both the robustness of our splicing calls and their biological coherence in the context of drug resistance mechanisms. Together, these findings solidify the validity of our splicing analysis framework.

Figure for referees not shown.

Response Figure 4. The robustness of splicing changes across all significant events. (A) Scatter plot showing the correlation between log₂FC of AKT2-206 isoforms and Delta-PSI of exon 10. (B) Cumulative curve distribution of Laplace entropy between genes that undergo differential splicing events and genes that do not undergo differential splicing events. Kolmogorov-Smirnov test. (C) The boxplot shows the Laplace entropy between genes that undergo differential splicing events and genes that do not undergo differential splicing events. Student's t-test. (D) The circular dendrogram shows KEGG pathways that are significantly enriched for genes undergoing differential splicing events. The size of circles represents the number of genes belonging to that pathway. (E) AKT2-206 expression changes in BRAF mutant melanoma datasets under five independent MAPKi studies.**** p < 0.0001.

Accordingly, we have made additional descriptions on pages 7, lines 125-128:

“And, genes harboring differential splicing events in the DP comparison group were strongly enriched in signal transduction pathways associated with MAPKi resistance,

including mTOR and PI3K-AKT signaling pathways (Appendix Fig. S1).”

And on pages 8, lines 136-138: “The isoform heterogeneity of genes harboring differential splicing events (identified by rMATS) is significantly higher than that of non-altered genes (Appendix Fig. S3).”

And on pages 12, lines 231-234: “Simultaneously, we collated transcriptome sequencing data from five independent studies (GSE75299, GSE203545, GSE285131, GSE103630, GSE186108), and the results showed that AKT2-206 expression was significantly increased in 57.7% (15/26) of acquired resistance samples (Appendix Fig. S9A).”

And on pages 13, lines 246-249: “Meanwhile, we found that there was a significant correlation between the log₂FC value of AKT2-206 and the Delta-Percent-Spliced-In (PSI) value of exon 10 ($R = 0.41$, p value = 0.038, Appendix Fig. S9B), indicating that changes in AKT2-206 expression are closely related to the splicing regulation of exon 10.”

Correspondingly, we have integrated this result into Appendix Fig. S1, S3, and S9.

3. A critical follow-up question is whether this AKT2 splicing event can be therapeutically targeted to enhance sensitivity to MAPK inhibitors. This could be experimentally tested using an antisense oligonucleotide (ASO) specifically targeting the AKT2 splicing event. Such experiments would also help quantify the direct contribution of this splicing switch to resistance.

Response: We sincerely appreciate the reviewer’s constructive suggestion regarding the therapeutic potential of targeting the AKT2 splicing event. In response to this important question, we designed an exon 10-specific ASO based on our previous finding that AKT2-206 isoform arises through exon 10 inclusion, while AKT2-210 results from exon 10 skipping in AKT2 pre-mRNA. Transfection of exon 10-targeting ASO into A375 and SK-MEL-8 melanoma cell lines effectively silences exon 10, as validated by RT-PCR (**Response Fig. 5A**). Critically, clonogenic assays demonstrated

that ASO-mediated exon 10 silencing significantly sensitized melanoma cells to PLX4032 (**Response Fig. 5B**). These results directly establish the therapeutic potential of targeting the AKT2 exon 10 to overcome MAPK inhibitor resistance, with ASO-based exon 10 modulation serving as an actionable strategy to enhance treatment efficacy.

Figure for referees not shown.

Response Figure 5. ASO-mediated exon 10 silencing increased the sensitivity of melanoma cells to BRAFi. (A) Efficiency of ASO-mediated exon 10 silencing. The data are shown as mean±SD (n=3, technical replicates); ANOVA. (B) Colony formation assay was performed on melanoma cells transfected with negative ASO control and exon 10 ASO with or without PLX4032 (0.01µM). **** p < 0.0001.

Accordingly, we have made additional descriptions on pages 14, lines 271-275:

“In addition, we designed an antisense oligonucleotide (ASO) specific for exon 10, and long-term proliferation assays showed that ASO-mediated silencing of exon 10 significantly increased the sensitivity of melanoma cells to PLX4032, which suggested the therapeutic potential of targeting AKT2 exon 10 to overcome MAPKi resistance (Appendix Fig. S11).”

Correspondingly, we have integrated this result into Appendix Fig. S11.

Addressing the above concerns-particularly by clarifying the functional impact of AKT2 splicing, improving the presentation of statistical data, and considering therapeutic targeting-would significantly enhance the manuscript.

Response to Referee #2:

In this interesting study, the authors investigated the role of alternative pre-mRNA splicing in drug resistance of melanoma to MAPK-targeted therapy. The authors first used RNA-seq analyses to identify a large scale of splicing alterations in MAPKi resistant melanoma and found that the alternatively splicing isoforms are enriched in the MAPK and PI3K-AKT pathways. Then the authors identified a functionally significant splicing isoform in a key member of the AKT family, AKT2. Alternative splicing of exon 10 of AKT2 leads to a switch from isoform AKT2-210 to AKT2-206 and introduces the activated loop (A-loop) to the functional protein, which activates the PI3K-AKT pathway and thereby confers MAPKi resistance. The authors further showed that the alternative splicing of AKT2 exon 10 is potentially directly regulated by RNA splicing factor HNRNPK.

The manuscript is well written. The experiments were well executed and the data clearly presented. This study is novel and reshapes our understanding of the mechanism of drug resistance in melanoma from gene expression changes to alternative splicing (without changes in overall expression levels). Therefore, this study is of major significance in that respect. I think the manuscript will be improved if the authors can address the following comments.

Response: We thank the reviewer for the diligent review of our manuscript, and appreciate the comments that “This study is novel and reshapes our understanding of the mechanism of drug resistance in melanoma from gene expression changes to alternative splicing (without changes in overall expression levels).” Following your constructive recommendations, we have systematically addressed each query through supplementary experimental validation, as documented in the point-to-point responses section below. The manuscript has been revised accordingly.

Specific comments:

1. Figures 6B and 6C, the authors showed that the expression of isoform AKT2-206

was reduced upon HNRNPK knockdown. However, it was not clear whether the downregulation of AKT2-206 was caused by alternative splicing of exon 10 of AKT2 or by reduction of the overall transcription of AKT2. The authors may want to perform an RT-PCR analysis to confirm that AKT2 exon 10 is indeed alternatively spliced upon HNRNPK knockdown. Specifically, the authors may perform an RT-PCR reaction (with a forward primer annealing to the complementary strand of exon 9 and a reverse primer annealing to exon 11) and obtain two PCR products of different sizes in the same reaction (a larger-sized PCR product with exon 10 included and a shorter-sized PCR product without exon 10). Then the authors can estimate the PSI (percent spliced in) of exon 10 by quantifying these two PCR products resolved in an agarose gel. The PSI value of exon 10 is expected to decrease upon HNRNPK knockdown if AKT2 exon 10 is indeed alternatively spliced.

Response: We sincerely appreciate your insightful guidance in clarifying the regulatory role of hnRNPK in AKT2 exon 10 splicing. As recommended, we performed additional experiments to rigorously evaluate whether hnRNPK knockdown directly induces alternative splicing of AKT2 exon 10. Specifically, primers flanking exons 9-11 of AKT2 were designed (**Response Table 1**), and splicing changes of exon 10 were analyzed by RT-PCR coupled with agarose gel electrophoresis following hnRNPK knockdown. The results unambiguously demonstrated (**Response Fig. 1**) that hnRNPK-knockdown cells exhibited an increase in the shorter PCR product (skipping exon 10) and a concurrent decrease in the longer isoform (including exon 10). Quantitative analysis revealed a reduction in the exon 10 PSI value, confirming that hnRNPK knockdown promotes exon 10 skipping. These findings definitively establish hnRNPK as a critical regulator of AKT2 exon 10 alternative splicing.

Figure for referees not shown.

Response Figure 1. qPCR analysis of AKT2 exon 10-inclusive and exon 10-skip isoforms before and after hnRNPK knockdown. Long-term exposure was used to observe changes in PCR products that do not contain exon 10, and short-term exposure was used to observe changes in PCR products that contain exon 10 (n=3, biological replicates).

Response Table 1. The qPCR primers for exons 9 to 11 of the AKT2 gene.

Target	Exons 9-11
Forward primer	CTCGGCTCTTGAGTACTTG
Reverse primer	CTGGAGGACAATGACTATGG

Accordingly, we have made additional descriptions on pages 16, lines 313-317:

“Notably, the reduction in AKT2-206 transcription was accompanied by a decrease in the PSI of exon 10, defined as the percentage of mRNA containing exon 10 relative to the total AKT2 mRNA. This suggests that hnRNPK may act as an RBP to promote the production of AKT2-206 by regulating the alternative splicing of exon 10 (Fig. 6D).”

Correspondingly, we have integrated this result into Fig. 6D.

2. The authors predicted seven HNRNPK binding sites in AKT2 transcripts (as shown in Table S5). Are any of these HNRNPK binding sites located in exon 10, intron 9 or intron 10 of AKT2? It would be helpful if the authors could provide a simple illustration of the locations of HNRNPK binding sites relative to the position of exon 10 of AKT2. If the HNRNPK binding sites are not located very close to the splice sites of exon 10, then the authors need to provide an explanation for how the alternative splicing of AKT2 exon 10 is directly regulated by HNRNPK.

Response: We appreciate the reviewer's careful analysis and insightful commentary regarding the locations of the hnRNPk binding sites in the AKT2 transcript and their potential influence on exon 10 splicing. Upon reviewing the locations of the seven predicted hnRNPk binding sites in AKT2 (**Response Fig. 2**), we confirm that these binding sites are not located in close proximity to the splice sites of exon 10, nor within exon 10, intron 9, or intron 10. We understand that this raises important questions about how hnRNPk could regulate the alternative splicing of exon 10 in AKT2.

Figure for referees not shown.

Response Figure 2. Location of the hnRNPk binding site relative to AKT2 exon 10.

While the hnRNPk binding sites are not immediately adjacent to the splice sites, it is important to note that hnRNPk is known to regulate splicing through mechanisms that extend beyond simple proximity to splice sites (Expert-Bezançon *et al*, 2002; Huang *et al*, 2024; Tyson-Capper & Gautrey, 2018). As a multifunctional RNA-binding protein, hnRNPk can regulate alternative splicing through long-range interactions, dynamic coordination with other splicing factors, and structural remodeling of pre-mRNA. For instance, hnRNPk modulates SYNGAP1 splicing by binding to its 3' UTR rather than near the alternative exons, demonstrating its ability to influence splicing distally (Araki *et al*, 2020; Yokoi *et al*, 2022). We propose that hnRNPk may regulate AKT2 exon 10 inclusion through multiple mechanisms: (1) remodeling RNA secondary/tertiary structure to alter splice site accessibility (Jones *et al*, 2022); (2) recruiting core splicing machinery or competing with repressive factors (Kong *et al*, 2024). For example, hnRNPk competes with CPSF6 for NUDT21 binding, blocking NEAT1_1 maturation while promoting NEAT1_2 production (Naganuma *et al*, 2012); (3) indirectly modulating splicing by affecting the expression or activity of other splicing regulators through transcriptional or post-transcriptional

regulatory mechanisms like SRSF1 (akin to its role in promoting oncogenic CD44 variants) (Peng *et al*, 2019); and (4) coupling with transcription dynamics to influence Pol II elongation rates and spliceosome recruitment timing (Ip *et al*, 2011; Mikula *et al*, 2013). These established mechanisms collectively support that hnRNPK's distal binding can still critically govern exon 10 splicing via integrated transcriptional, structural, and competitive regulatory networks. We fully agree that it is critical to clarify the mechanisms by which hnRNPK regulates the splicing of AKT2 and plan to prioritize these studies in subsequent work. For now, our data provide compelling evidence that hnRNPK acts as a key regulator of AKT2 splicing switch, functionally linking RNA-binding protein dysregulation to MAPKi resistance—a conclusion robustly supported by functional rescue experiments in this manuscript.

Accordingly, we have made additional descriptions on pages 20, lines 386-395:

“The alternative splicing of AKT2 exon 10 was regulated by a spliceosome complex that contained hnRNPK. Although the binding sites of hnRNPK to AKT2 are distal to introns 9-10 or exon 10 of AKT2 (Appendix Fig. S17), as a multifunctional RNA-binding protein, hnRNPK may still regulate alternative splicing of exon 10 through mechanisms such as remodeling of RNA secondary/tertiary structures (Jones *et al*, 2022), recruiting core splicing machinery or competing with repressive factors (Kong *et al*, 2024; Naganuma *et al*, 2012), indirectly modulating splicing by affecting the expression or activity of other splicing regulators (Peng *et al*, 2019), and coupling with transcriptional dynamics (Ip *et al*, 2011; Mikula *et al*, 2013).”

Correspondingly, we have integrated this result into Appendix Fig. S17.

3. Alternative splicing is species-specific. The majority of alternative splicing events is not conserved between human and mouse. The authors did not provide any evidence that the mechanism of alternative splicing of human AKT2 exon 10 is conserved in mouse Akt2 exon 10. In fact, according to Figures S11B and S11C, the switch between isoforms Akt2-210 and Akt2-201 is not caused by alternative splicing of exon 10 of mouse Akt2. The authors need to identify the new alternative splicing pattern that leads to the switch between isoforms Akt2-210 and Akt2-201 and then

show experimental evidence (e.g., RT-PCR results). Are there HNRNPK binding site(s) in mouse Akt2 transcripts that are potentially responsible for this new alternative splicing event?

Response: Thank you for raising this critical question regarding species-specific alternative splicing mechanisms and their implications for our comparative analysis. We fully agree that alternative splicing events are often not conserved across species, and we appreciate the opportunity to clarify both the rationale and limitations of our cross-species approach.

Our primary objective in analyzing mouse Akt2 isoforms (Akt2-210 and Akt2-201) was to investigate whether functional modulation of the A-loop (a key regulatory motif essential for AKT2 kinase activation) is conserved during therapy resistance, rather than asserting conservation of specific splicing mechanisms. It is indeed as you said that the conversion of the two isoforms of Akt2-210 and Akt2-201 is not caused by the alternative splicing of mouse Akt2 exon 10 (**Response Fig. 3A**). By analyzing the exon structure of Akt2-210 and Akt2-201, we found that there was mutual exclusion of exon 1 at the 5' end of Akt2-210 and Akt2-201, while there was retention of exons 5-14 at the 3' end (**Response Fig. 3A**). The RNA-seq data (**Appendix Fig. S15B, D**) reveal that Akt2-210 (lacking the A-loop) is downregulated, while Akt2-201 (retaining an intact A-loop) is upregulated in resistant mouse tumors. Critically, this isoform switch is accompanied by elevated protein levels of hnRNPk, p-AKT, and p-S6 in mouse-resistant samples, as demonstrated by IHC (**Figs. 6F, G**). This triad of elevated hnRNPk, activated AKT signaling, and downstream S6 phosphorylation in mouse recapitulates the behavior of human MAPKi-resistant melanoma samples, reinforcing the activation of A-loop-dependent kinase signaling across species. While the exact splice events driving the isoform switch differ between humans and mice, our transcript-level RNA-seq quantification and IHC results robustly support a conserved functional trend: depletion of A-loop-deficient isoforms and reactivation of AKT signaling. Together, these data underscore that divergent splicing mechanisms may converge on similar functional consequences, with hnRNPk upregulation and

A-loop restoration driving AKT/S6 activation in both species.

Although we do not assert direct mechanistic conservation of hnRNPK-mediated splicing regulation in mice, bioinformatic analysis (via POSTAR3, **Response Table 2**) identifies Hnrnpk as a potential regulator of Akt2 splicing. Furthermore, *in vitro* experiments demonstrate that hnRNPK knockdown in SMM102 cells significantly reduces Akt2-201 mRNA levels (**Response Fig. 3B**). Additionally, IHC data from drug-resistant mouse tumors reveal a correlation between hnRNPK protein levels and phosphorylated AKT and p-S6 (**Figs. 6F, G**). These results aligns with human data, where hnRNPK promotes A-loop restoration and AKT activation. Though species-specific splicing events likely govern splicing, the functional role of hnRNPK in A-loop restoration and pathway activation appears to be conserved.

We acknowledge that the regulators governing Akt2 splicing in mice remain unresolved. For instance, does hnRNPK directly bind mouse Akt2 transcripts to modulate splicing, or does it act indirectly via compensatory pathways? While our study prioritized establishing functional parallels in A-loop restoring and kinase activating, we have revised the Discussion to explicitly highlight this open question.

Figure for referees not shown.

Response Figure 3. Alternative splicing of Akt2 isoforms. (A) Exon structure of Akt2-210 and Akt2-201 splicing isoforms. (B) The expression levels of hnRNPK and Akt2-201 after knockdown of hnRNPK. The data are shown as mean±SD (n=3, technical replicates); ANOVA. **** p < 0.0001.

Response Table 2. POSTAR3 predicted binding sites between Hnrnpk and Akt2.

Splicing factor	Position	PhastCons score	PhyloP score	Method
-----------------	----------	-----------------	--------------	--------

Hnrnpk	chr7:27614200-27614250	0.99	1.804	DeepBind
Hnrnpk	chr7:27639350-27639400	0.398	0.387	DeepBind
Hnrnpk	chr7:27619200-27619250	0.052	-0.077	DeepBind
Hnrnpk	chr7:27604100-27604150	0.013	0.1	DeepBind
Hnrnpk	chr7:27599550-27599600	0.008	0.174	DeepBind
Hnrnpk	chr7:27612100-27612150	0.008	0.019	DeepBind
Hnrnpk	chr7:27632500-27632550	0.005	-0.386	DeepBind
Hnrnpk	chr7:27609950-27610000	0.003	-0.246	DeepBind
Hnrnpk	chr7:27594650-27594700	0.001	-0.313	DeepBind
Hnrnpk	chr7:27611300-27611350	0.001	-0.264	DeepBind
Hnrnpk	chr7:27639450-27639500	0.126	0.104	FIMO
Hnrnpk	chr7:27621450-27621500	0.081	0.128	FIMO
Hnrnpk	chr7:27604050-27604100	0.015	0.149	FIMO
Hnrnpk	chr7:27614200-27614250	0.99	1.804	TESS
Hnrnpk	chr7:27639350-27639400	0.398	0.387	TESS
Hnrnpk	chr7:27619200-27619250	0.052	-0.077	TESS
Hnrnpk	chr7:27604100-27604150	0.013	0.1	TESS
Hnrnpk	chr7:27599550-27599600	0.008	0.174	TESS
Hnrnpk	chr7:27612100-27612150	0.008	0.019	TESS
Hnrnpk	chr7:27632500-27632550	0.005	-0.386	TESS
Hnrnpk	chr7:27609950-27610000	0.003	-0.246	TESS

Hnrnpk	chr7:27594650-2759470 0	0.001	-0.313	TESS
Hnrnpk	chr7:27611300-2761135 0	0.001	-0.264	TESS

Accordingly, we have made additional descriptions on pages 17-18, lines 334-360:

“All Akt2 isoforms expressed in the tumors were identified, with structural domain annotation revealing that mouse Akt2-210 and Akt2-201 respectively lacked or retained the A-loop catalytic domain (Appendix Fig. S15B). Strikingly, despite differences in exon usage between species (Appendix Fig. S15C), mouse tumor samples also showed conserved depletion of A-loop-deficient isoforms (Akt2-210) and enrichment of A-loop-intact isoforms (Akt2-201) during resistance (Appendix Fig. S15D). Crucially, this splicing switch converged on activation of AKT signaling characterized by elevated p-AKT and p-S6 in resistant tumors (Fig. 6F, G), mirroring what was observed in MAPKi-resistant melanoma cell lines. Mechanistically, while the precise splicing elements driving the splicing switch may differ between species, the hnRNPK elevation observed in resistant mouse tumors correlated strongly with both Akt imbalance and PI3K-AKT pathway activation (Fig. 6F, G). This functional triad - hnRNPK upregulation, A-loop restoration via splicing, and AKT/S6 phosphorylation - bridges species-specific transcript diversity to conserved signaling adaptations. Bioinformatic analysis using POSTAR3 (Zhao et al, 2022) further predicted potential Hnrnpk binding motifs in mouse Akt2 splicing regulatory regions (Table EV6). Furthermore, in vitro experiments confirmed that knockdown of Hnrnpk in SMM102 cells significantly reduced Akt2-201 mRNA levels (Appendix Fig. S15E). Importantly, these preclinical findings align with clinical evidence showing HNRNPK elevation in 75% of progressed patient biopsies (3/8 with dual elevation of HNRNPK and p-AKT, Fig. 6H, Table EV7). These parallel observations across species reinforce that hnRNPK-mediated alternative splicing serves as a conserved strategy to reactivate A-loop-dependent AKT signaling during MAPKi resistance, despite species-specific splicing patterns. Collectively, our results demonstrate that the hnRNPK-regulated splicing program represents a novel mechanism driving MAPK

pathway inhibitor resistance, providing translational insights for targeting splicing-mediated kinase rewiring in refractory melanoma.”

And on pages 20-21, lines 398-407: “Consistently, the Akt2 splicing switch from Akt2-210 to Akt2-201 in MAPKi-treated mouse model recapitulates the splicing pattern mediating A-loop restoration observed in melanoma patients. Notably, this splicing switch in mice involves different splicing events that are different from those in melanoma patients. Despite species differences in splicing mechanisms, this switch is still significantly associated with hnRNPK upregulation, A-loop recovery, and PI3K-AKT pathway activation. Importantly, preclinical findings align with clinical data showing HNRNPK/p-AKT co-elevation in 37.5% of progressing patients (n=8), underscoring the conservation of hnRNPK-mediated A-loop-dependent AKT kinase hyperactivity in MAPKi resistance despite interspecies splicing diversity.”

Correspondingly, we have integrated this result into Appendix Fig. S15C, E, and Dataset EV6.

4. *Figure S11C, it would be helpful if the authors could provide the sequences of the qPCR primers used for detection of the expression of mouse Akt2 isoforms Akt2-210, Akt2-201, Akt2-205 and Akt2-214.*

Response: Regarding your inquiry about the qPCR primers for detecting mouse Akt2 isoforms (Akt2-210, Akt2-201, Akt2-205, and Akt2-214) in **Appendix Fig. S15D**, we would like to clarify that the results presented in this figure were derived from RNA sequencing data of mouse tumor tissues and did not involve qPCR experiments. Therefore, no qPCR primers were utilized for this specific analysis.

Nonetheless, we fully appreciate your suggestion to provide the primer sequences for completeness. The sequences of the qPCR primers designed for these Akt2 isoforms are below (**Response Table 3**, or as **Reagents and Tools Table**) for your reference:

Response Table 3. The qPCR primers of mouse Akt2 isoforms Akt2-210, Akt2-201, Akt2-205 and Akt2-214.

Target gene	Forward primer	Reverse primer
Akt2-201	GCTGAAAGGAGACTGTAAAA	TACAGTATCGTCTTGGGTCT
Akt2-205	TAAACAGCTTCTGTGTGGAA	GCAGCTGTAATCAAATGTCC
Akt2-210	ATGCAGAGGTCTTGGAAAAG	ATACAGTATCGTTGCACACA
Akt2-214	ATGTCTCTCCCTCCTCAATA	AGTAAGGAAAGGCCAGCTAT

5. Figure 1A, the pie chart of (Melanocyte vs. Nevus) is 100% identical to that of (Nevus vs. Melanoma). The authors need to provide an explanation for this.

Response: We sincerely thank the reviewer for identifying this critical issue in Figure 1A. After a thorough investigation, we confirm this resulted from an inadvertent error during the multi-panel figure compilation process. Specifically, the pie chart of "Nevus vs. Melanoma" erroneously replaced the pie chart of "Melanocyte vs. Nevus" during panel organization. We unequivocally confirm this was a graphical processing error, and the underlying experimental data and statistical analyses remain valid. The revised Fig. 1A now accurately displays the distinct splicing profiles for all comparisons (**Response Fig. 4**). We deeply apologize for any confusion caused.

Figure for referees not shown.

Response Figure 4. Characteristics of differential splicing events at various stages of melanoma development and drug resistance evolution. The pie charts illustrate the distribution of various splicing event modes. Different colors indicate distinct modes of splicing events.

6. It would be helpful if the authors could specifically mention in the first paragraph of the Results section that the data shown in Figures 1A and 1B were obtained by using the alternative splicing detection tool, rMATS.

Response: Thank you for your suggestion. We re-wrote the Results section of the

manuscript and made the following revisions.

We made corrections on pages 7, line 118-121: “We first characterized five canonical alternative splicing pattern by rMATS, namely exon skipping/inclusion (SE), alternative 5' splice-site selection (A5SS), alternative 3' splice-site selection (A3SS), intron retention (RI), and mutually exclusive exons (MXE).”

7. Figure 1B, it would be helpful if the authors could clarify A3SS-skip vs. A3SS-inclusive, A5SS-Skip vs. A5SS-inclusive, and MXE-Skip vs. MXE-inclusive. Specifically, does A3SS-Skip mean skipping of the proximal 3' splice site or skipping of the distal 3' splice site? Does A5SS-Skip mean skipping of the proximal 5' splice site or skipping of the distal 5' splice site? Does MXE-Skip mean skipping of the upstream exon or skipping of downstream exon?

Response: Thank you for your suggestion. A3SS-Skip means skipping of the proximal 3' splice site. A5SS-Skip means skipping of the distal 5' splice site. MXE-Skip means skipping of the upstream exon (**Response Fig. 5**). We have supplemented the description of A3SS-skip, A3SS-inclusive, A5SS-Skip, A5SS-inclusive, MXE-Skip, and MXE-inclusive in the legend of Fig. 1B of the revised manuscript.

Figure for referees not shown.

Response Figure 5. Five alternative splicing events detected by rMATS (NIH, 2025).

Response to Referee #3:

Manuscript: EMBOR-2024-61082V1

Title: "Transcriptome-wide decoding the roles of aberrant splicing in melanoma MAPK-targeted resistance evolution"

The authors assess comprehensively the magnitude of alternative splicing events during melanoma development/progression and MAPKi treatment (targeted therapy) by re-analyzing already published RNA expression data of melanocytes, nevi, and melanomas (baseline vs. targeted therapy). They apply a computational decision tree, allowing the identification of functional splicing isoforms through pathway enrichment analysis while discarding principal isoforms and focusing on repeatedly up/downregulated ones in MAPKi resistant melanoma samples. As probably expected, the MAPK pathway gene set seems to be the most perturbed at the splicing level, but so appears the PI3K-AKT pathway. The authors show evidence for a splicing switch from AKT2 isoform 210 to AKT2 isoform 206 which results in kinase hyperactivity (increase pAKT (S473)) through restoration of the activated fragment A-loop. Functional validation via ectopic overexpression of AKT2 206 and 210 in melanoma cell lines and Xenograft models underlines the role of the isoform AKT2 206 in conferring resistance to BRAF inhibition. Finally, the authors present evidence that the splicing switch from AKT 210 to 206 is mediated by hnRNPK as shRNA-mediated KD of hnRNPK attenuated AKT2-206 expression in parental and resistant SK-MEL-28 and A375 melanoma cells and resensitized them to BRAFi. Lastly, IHC evidence is presented in murine (+/-BRAFi) and human melanomas (baseline vs. DP) showing a co-expression pattern of hnRNPK, pAKT and pS6.

In summary, the manuscript provides solid evidence that the novel splicing switch from AKT2-210 to AKT2-206 confers resistance to BRAFi.

Response: We sincerely appreciate the reviewer's thorough evaluation and positive

assessment of our work. Please see below for our point-by-point response to your suggested revisions, and we hope that the manuscript has been adequately revised.

The following points should be addressed to further strengthen the manuscript:

Figure 1A) Piecharts of Melanocyte vs. Nevus and Nevus vs. Melanocytes are redundant, right?

Response: We sincerely thank the reviewer for identifying this critical issue in Figure 1A. After a thorough investigation, we confirm this resulted from an inadvertent error during the multi-panel figure compilation process. Specifically, the pie chart of "Nevus vs. Melanoma" erroneously replaced the pie chart of "Melanocyte vs. Nevus" during panel organization. We unequivocally confirm this was a graphical processing error, and the underlying experimental data and statistical analyses remain valid. The revised Fig. 1A now accurately displays the distinct splicing profiles for all comparisons (**Response Fig. 1**). We deeply apologize for any confusion caused.

Figure for referees not shown.

Response Figure 1. Characteristics of differential splicing events at various stages of melanoma development and drug resistance evolution. The pie charts illustrate the distribution of various splicing event modes. Different colors indicate distinct modes of splicing events.

Figure 2A) The middle section of the alluvial plots reads: "Domain retainion"? Do you mean "domain retention"?

Response: We are grateful to the reviewer for identifying the typographical error in Fig. 2A, where "Domain retain" was inadvertently misspelled as "Domain retainion".

We sincerely apologize for this oversight and value the reviewer's meticulous attention to detail, which has strengthened the clarity of our scientific presentation.

Figure 2C) The PI3K-AKT pathway has the highest enrichment count (number of DEIs) but a relatively poor p-value, could you comment on that?

Response: We thank the reviewer for this insightful inquiry regarding the relationship between enrichment count and statistical significance in Fig. 2C. Gene list enrichment is based on the over-representation analysis (ORA) method, a simple and frequently used gene set enrichment method, ORA calculates enrichment significance through a hypergeometric test, where p-value sensitivity is inversely correlated with pathway gene set size. This observation directly reflects the mathematical properties of ORA and the biological characteristics of the pathways analyzed.

The hypergeometric test calculates the probability of observing K differentially expressed isoform (DEI) -corresponding genes in a pathway of size M against a background of N genes:

$$P = \sum_{i=k}^n \frac{\binom{M}{i} \binom{N-M}{n-i}}{\binom{N}{n}}$$

Where:

N : Total background genes

M : Genes in target pathway (PI3K-AKT: 354 vs MAPK: 295)

n : DEI-corresponding genes

K : Genes corresponding DEIs overlapping with pathway

Despite higher absolute DEI-corresponding gene counts, the larger PI3K-AKT pathway inherently dilutes statistical "surprise" because the null hypothesis assumes proportional distribution of DEI-corresponding genes across all pathways.

To ensure transparency and address potential reader confusion, we have now explicitly incorporated this statistical rationale into the Methods section of the revised manuscript.

We made corrections on pages 30, line 511-514: "Pathway enrichment analysis was

conducted using KOBAS (FDR < 0.05) (Xie et al., 2011), statistical significance in KOBAS was calculated via over-representation analysis using the hypergeometric test.”

Figure 3A,B) What depicts the X axis? Patient samples, if yes, what cohort? doi: 10.1016/j.cell.2015.07.061? Also 3E is this from the same patient cohort?

If yes, could you check if AKT2-206 overexpression is mutually exclusive with genetically driven MAPKi resistance mechanisms?

Regarding the frequency of AKT2-206 expression, could you mine alternative melanoma data sets under MAPKi? Or how about mining the CCLE database as it holds RNA and drug response data? Would you expect to find the splicing switch also in NRAS mutant melanoma challenged with MEKi?

Response: Thank you for raising this critical question. The X-axis in Fig. 3A, B represents patient samples from the GSE65186 (doi: 10.1016/j.cell.2015.07.061) and EGA S00001000992 (doi:10.1172/JCI78954) cohorts. These samples were derived from melanoma patients treated with MAPK inhibitor regimens (BRAFi or BRAFi + MEKi), including matched pre-treatment baseline and progressive disease tumor samples.

To determine whether AKT2-206 overexpression is mutually exclusive with genetically driven MAPKi resistance mechanisms (**Response Fig. 2A**), we performed co-occurrence analysis using Cramer's V correlation and Fisher's exact test. The results demonstrated no significant association between AKT2-206 overexpression and previously identified MAPKi resistance mechanisms (genomic, transcriptomic, or epigenetic drivers), as evidenced by Fisher's exact test yielding non-significant p-values ($p > 0.05$) in all comparisons (**Response Fig. 2B**). These findings suggest that AKT2-206 overexpression operates independently of canonical MAPKi resistance mechanisms rather than exhibiting mutual exclusivity or concomitant effects.

Figure for referees not shown.

Response Figure 2 Independence of AKT2 splicing switch from known resistance mechanisms. (A) Recurrence of alterations and heterogeneity mechanistic of known resistance genes and AKT2-206 in acquired resistant tumors. (B) Correlation analysis of AKT2-206, *BRAF*^{V600E} alternative splicing, PIK3CA and identified drug resistance drivers.

We appreciate the reviewer's insightful suggestion to explore additional datasets (such as CCLE) for AKT2-206 expression analysis. While the CCLE database provides RNA-seq and drug response data from cancer cell lines, it does not include longitudinal sequencing data comparing treatment-naïve vs. MAPKi-resistant counterparts, which is essential for examining the dynamic changes in AKT2-206 expression upon resistance development. To address this limitation, we rigorously screened public repositories (e.g., GEO, ENA) and compiled transcriptomic

sequencing data from five independent studies (GSE75299, GSE203545, GSE285131, GSE103630, GSE186108), encompassing 26 matched baseline and post-MAPKi-resistant samples. Strikingly, AKT2-206 expression was elevated (FC > 1.2) in 57.7% (15/26) of acquired resistance samples, strongly supporting its association with MAPKi resistance (**Response Fig. 3A**).

Figure for referees not shown.

Response Figure 3 Expression patterns of AKT2-206. (A and B) AKT2-206 expression changes in BRAF mutant melanoma datasets under MAPKi (A), NRAS mutant melanoma PDX models under MEKi (B). (C) AKT1-218 expression changes in NRAS mutant melanoma PDX models under MEKi. Upregulation and downregulation patterns are displayed in red and blue, respectively.

While MEK inhibitors (e.g., trametinib) are not standard therapies for NRAS-mutant melanoma in clinical practice, we sought to leverage the publicly available dataset GSE158607 to investigate whether MEK inhibition induces similar splicing switches in this subtype. This dataset comprises paired pre-treatment and acquired trametinib-resistant samples from NRAS mutant melanoma patient-derived xenograft (PDX) models (n=5 baseline, n=16 resistant). Intriguingly, our analysis revealed an inverse trend compared to BRAFi-resistant melanomas, AKT2-206 was downregulated in MEKi-resistant tumors (**Response Fig. 3B**). However, we found that the main transcript of AKT1 gene, AKT1-218, was upregulated in multiple pairs of drug-resistant samples. (**Response Fig. 3C**). This suggests that MEKi resistance in NRAS-mutant melanomas may involve splicing switches of other genes in the AKT

family.

However, we emphasize that these observations are based on a limited sample size, particularly at baseline (n=5), which reduces statistical power and increases the risk of overinterpreting stochastic variability. PDX models, while valuable, may not fully recapitulate the tumor microenvironment or resistance mechanisms seen in human patients. Thus, while the dataset provides a preliminary signal, the findings require validation in larger, well-powered cohorts to confirm their biological relevance.

Accordingly, we have made additional descriptions on pages 19-20, lines 378-386:

“Furthermore, we correlated AKT2-206 splicing switching with known MAPKi resistance drivers (e.g., NRAS mutations, BRAF splice variants, NF1 loss) (Di Leo et al, 2024; Huang et al, 2023; Hugo et al., 2015; Shao & Aplin, 2010). Results showed no significant association between AKT2-206 upregulation and known genomic, transcriptional, or epigenetic resistance mechanisms ($p > 0.05$, Appendix Fig. S16A, B). Strikingly, 5.00% (2/40) of advanced melanomas exhibited upregulated AKT2-206 expression in the absence of established resistance drivers, suggesting that it may serve as an independent resistance mechanism in molecularly distinct patient subgroups.”

And on pages 12, lines 231-234: “Simultaneously, we collated transcriptome sequencing data from five independent studies (GSE75299, GSE203545, GSE285131, GSE103630, GSE186108), and the results showed that AKT2-206 expression was significantly increased in 57.7% (15/26) of acquired resistance samples (Appendix Fig. S9A).”

Correspondingly, we have integrated this result into Appendix Fig. S9A, S16A, B.

General comment/question: Are both AKT2 isoforms (-210 and -206) equally sensitive to pan-AKTi or AKT2 inhibitors?

Response: Thank you for raising this important question. Our experimental results demonstrate that AKT2 isoforms exhibit different drug sensitivities. In two melanoma cell lines (A375 and SK-MEL-28), the AKT2-210 overexpression cells showed

increased sensitivity to both pan-AKT inhibitors (pan-AKTi, capivasertib) and AKT2-specific inhibitors (AKT2i, CCT128930) compared to AKT2-206 overexpression cells. At high inhibitor concentrations (100 μ M), both AKT2-206 and AKT2-210 overexpression cells exhibited increased sensitivity to AKT2i. At low concentrations (\leq 10 μ M), AKT2-206 overexpression cells were equally sensitive to pan-AKTi and AKT2i in A375 cells, while AKT2-210 overexpression cells were significantly more sensitive to pan-AKTi; AKT2-206 overexpression cells were more sensitive to AKT2i, while AKT2-210 overexpression cells were still more sensitive to pan-AKTi in SK-MEL-28 cells (**Response Fig. 4**).

Figure for referees not shown.

Response Figure 4 Cell viability assays in melanoma cells with AKT2-206 and AKT2-210 overexpression after AKT2-specific inhibitor CCT128930 and pan-AKT inhibitor capivasertib treatment. The data are shown as mean \pm SD (n=5, biological replicates)

Accordingly, we have made additional descriptions on pages 15, lines 280-284:

“And the results of short-term proliferation assays showed that the AKT2-210 overexpression cells showed increased sensitivity to both pan-AKT inhibitors (pan-AKTi, capivasertib) and AKT2-specific inhibitors (AKT2i, CCT128930) compared to AKT2-206 overexpression cells in A375 and SK-MEL-28 cell line (Appendix Fig. S12A).”

Correspondingly, we have integrated this result into Appendix Fig. S12A.

Page 10, Line178: "In addition, 11 splicing isoforms corresponding to 10 genes also presented aberrant expression in SKCM cohort from TCGA, suggesting their fundamental roles in tumor progression and drug resistance." I am not sure how far TCGA_SKCM AS-analysis can suggest this as there aren't any drug-treated samples included to my knowledge. What could be compared in TCGA_SKCM is OS, primary vs. metastatic melanoma.

Response: Thank you for raising this critical point. We sincerely apologize for the lack of clarity in our original manuscript, which may have led to a misunderstanding of the intended logic. Our goal was to highlight two distinct but complementary lines of evidence supporting the functional relevance of 43 genes identified as experiencing isoforms transcriptional changes in the PI3K-AKT pathway. First, 31 of these genes were identified through mRNA expression profiling tied to drug-sensitivity data from the RNAactDrug database, suggesting their potential role in drug resistance-driven adaptation. Second, 11 splicing isoforms corresponding to 10 genes presenting aberrant expression in the SKCM cohort from TCGA were proposed as a separate observation, implying their possible involvement in tumorigenesis. Importantly, we did not intend to directly link the aberrant changes in TCGA-SKCM data to drug resistance, as we fully acknowledge that TCGA lacks drug treatment information. Instead, we aimed to emphasize that these genes may have dual functional roles: both in tumor progression (supported by their aberrant expression in TCGA) and in drug resistance (supported by RNAactDrug drug sensitivity data). We agree that the wording of the original paper could have misled readers, and we have revised the concluding statement to reflect this distinction.

We made corrections on pages 11-12, line 213-218: "RNAactDrug database revealed 31 genes functionally tied to drug sensitivity (Dong et al, 2020), while 11 splicing isoforms corresponding to 10 genes also presented aberrant expression in SKCM cohort from TCGA, implicating these genes in disease progression.

Collectively, these findings highlight a possible dual biological mechanism for the 43 genes identified above, which may be linked to both drug resistance-driven adaptation and tumorigenesis.”

Discussion:

When discussing the role of aberrant splicing in melanoma as means to confer resistance to MAPKi, the authors should include the paper doi.org/10.1172/JCI82534, where authors could show in multiple human melanoma cell lines and PDX mouse models, ASO-mediated skipping of exon 6 decreased MDM4 abundance, inhibited melanoma growth, and enhanced sensitivity to MAPKi.

Response: We sincerely thank the reviewer for highlighting the seminal work by Dewaele et al. (doi.org/10.1172/JCI82534), which elegantly demonstrates that ASO-mediated skipping of MDM4 exon 6 re-sensitizes melanomas to MAPK inhibition. We have now explicitly integrated this critical reference into our Discussion section.

We made corrections on pages 21, line 411-414: “While these studies defined canonical resistance mechanisms, emerging evidence underscores alternative splicing as a pervasive adaptive strategy enabling malignant escape from therapeutic pressure (Aya et al, 2024; Dewaele et al, 2016). However, prior studies focused on single-gene splicing events.”

References

- Araki Y, Hong I, Gamache TR, Ju S, Collado-Torres L, Shin JH, Haganir RL (2020) SynGAP isoforms differentially regulate synaptic plasticity and dendritic development. *Elife* 9
- Di Leo L, Pagliuca C, Kishk A, Rizza S, Tsiavou C, Pecorari C, Dahl C, Pacheco MP, Tholstrup R, Brewer JR *et al* (2024) AMBRA1 levels predict resistance to MAPK inhibitors in melanoma. *Proc Natl Acad Sci U S A* 121: e2400566121
- Expert-Bezançon A, Le Caer JP, Marie J (2002) Heterogeneous nuclear ribonucleoprotein (hnRNP) K is a component of an intronic splicing enhancer complex that activates the splicing of the alternative exon 6A from chicken beta-tropomyosin pre-mRNA. *J Biol Chem* 277: 16614-16623
- Huang F, Cai F, Dahabieh MS, Gunawardena K, Talebi A, Dehairs J, El-Turk F, Park JY, Li M, Goncalves C *et al* (2023) Peroxisome disruption alters lipid metabolism and potentiates antitumor response with MAPK-targeted therapy in melanoma. *J Clin Invest* 133
- Huang Y, Liu Y, Pu M, Zhang Y, Cao Q, Li S, Wei Y, Hou L (2024) SOX2 interacts with hnRNPK to modulate alternative splicing in mouse embryonic stem cells. *Cell Biosci* 14: 102
- Hugo W, Shi H, Sun L, Piva M, Song C, Kong X, Moriceau G, Hong A, Dahlman KB, Johnson DB *et al* (2015) Non-genomic and Immune Evolution of Melanoma Acquiring MAPKi Resistance. *Cell* 162: 1271-1285
- Ip JY, Schmidt D, Pan Q, Ramani AK, Fraser AG, Odom DT, Blencowe BJ (2011) Global impact of RNA polymerase II elongation inhibition on alternative splicing regulation. *Genome Res* 21: 390-401
- Jones AN, Graß C, Meininger I, Geerlof A, Klostermann M, Zarnack K, Krappmann D, Sattler M (2022) Modulation of pre-mRNA structure by hnRNP proteins regulates alternative splicing of MALT1. *Science Advances* 8: eabp9153
- Kong Y-Y, Shu W-J, Wang S, Yin Z-H, Duan H, Li K, Du H-N (2024) The methyltransferase SETD3 regulates mRNA alternative splicing through interacting with hnRNPK. *Cell Insight* 3: 100198
- Mikula M, Bomsztyk K, Goryca K, Chojnowski K, Ostrowski J (2013) Heterogeneous Nuclear Ribonucleoprotein (HnRNP) K Genome-wide Binding Survey Reveals Its Role in Regulating 3'-end RNA Processing and Transcription Termination at the Early Growth Response 1 (EGR1) Gene through XRN2 Exonuclease. *Journal of Biological Chemistry* 288: 24788-24798
- Naganuma T, Nakagawa S, Tanigawa A, Sasaki YF, Goshima N, Hirose T (2012) Alternative 3'-end processing of long noncoding RNA initiates construction of nuclear paraspeckles. *Embo j* 31: 4020-4034
- NIH, 2025. <https://bioinformatics.ccr.cancer.gov/docs/btep-coding-club/CC2023/rmats/>.
- Peng WZ, Liu JX, Li CF, Ma R, Jie JZ (2019) hnRNPK promotes gastric tumorigenesis through regulating CD44E alternative splicing. *Cancer Cell Int* 19: 335
- Shao Y, Aplin AE (2010) Akt3-mediated resistance to apoptosis in B-RAF-targeted melanoma cells. *Cancer Res* 70: 6670-6681
- Tyson-Capper A, Gautrey H (2018) Regulation of Mcl-1 alternative splicing by hnRNP F, H1 and K in breast cancer cells. *RNA Biol* 15: 1448-1457
- Yokoi S, Ito T, Sahashi K, Nakatochi M, Nakamura R, Tohnai G, Fujioka Y, Ishigaki S, Udagawa T, Izumi Y *et al* (2022) The SYNGAP1 3'UTR Variant in ALS Patients Causes Aberrant SYNGAP1 Splicing and Dendritic Spine Loss by Recruiting HNRNPK. *J Neurosci* 42: 8881-8896

Dear Prof. Shi,

Thank you for the submission of your revised manuscript. We have now received the enclosed reports from the referees and I am happy to say that all support its publication now. The referees still have a few more minor suggestions that I would like you to incorporate before we can proceed with the official acceptance of your manuscript.

A few editorial requests will also need to be addressed:

- Please move the Data Availability Section to before the Acknowledgments.
- Please remove the author credits from the ms file. All credits need to be entered during online ms submission.
- There are 7 EV tables uploaded in one Excel file. EV tables need to be uploaded as separate, individual files. Table EV1 and Table EV5 are Datasets so they need to be renamed to Dataset EV1 and Dataset EV2 in all places (source file name, title in ms submission system, legend, ms callout). The remaining EV tables should be uploaded as separate files called Table EV1-Table EV5 and their names need to be corrected in all files and ms callouts.
- The APPENDIX FILE is good but the page numbers need to be inserted throughout the file to number all pages.
- The Reagents & Tools TABLE needs to be removed from the ms and uploaded as a separate file.
- Materials and Methods should be just Methods.

Figure Legends - Comments

- Please note that the exact p values are not provided in the legends of figures 3E, 5A, C, D; 6B, C, D, E, G. Exact values need to be provided as reasonable.
- Please indicate the statistical test used for data analysis in the legend of figure 2C
- Please note that the box plots need to be defined in terms of minima, maxima, centre, bounds of box and whiskers, and percentile in the legends of figures 1C, 3E, F
- Please note that information related to n is missing in the legends of figures 3C, E, F
- Please note that the error bars are not defined in the legend of figure 3C.

I would like to suggest some minor changes to the abstract that needs to be written in present tense. Do you agree with this:

Drug resistance critically limits the long-term efficacy of MAPK-targeted therapy in melanoma. While resistance mechanisms at genetic, epigenetic, and transcriptional scales are well-documented, post-transcriptional splicing regulation remains poorly understood. By analyzing patient-matched pre-treatment and resistant melanoma biopsies, we uncover widespread alternative splicing alterations during therapy resistance. Splicing perturbations are most pronounced in MAPK and PI3K-AKT pathway genes. We identify a splicing switch of AKT2 from isoform 210 to 206 in 29.55% (13/44) of disease-progressive biopsies. This splicing switch induces AKT2 kinase hyperactivity by restoring the activated fragment A-loop. Functional validations confirm that AKT2-206 confers BRAF inhibitor resistance in melanoma cells by activating S6 kinase. Further, the splicing factor hnRNPk likely drives the splicing switch of AKT2 during acquired resistance. Our results not only provide insights into splicing-mediated regulation of drug resistance but also highlight the importance of alternative splicing isoforms as targets for clinical diagnosis and therapy.

Referee #1:

The authors have adequately addressed the concerns raised. However, the delta-PSI values presented on the y-axis of Response Fig. 4A appear modest, suggesting that alternative splicing changes in AKT2 alone may not fully explain the observed cellular phenotype. As such, we recommend that any claims regarding the role of AKT2 splicing be appropriately tempered in the manuscript.

Referee #2:

In this revision, the authors have adequately addressed all of my questions. The conclusions of this study have been strengthened, and the revised manuscript has been greatly improved. However, there are still minor points that the authors need to address.

Minor points:

1. Figure 6D shows images of RT-PCR products run on agarose gels, but the authors did not provide the RT-PCR method in the Materials and Methods section.

2. Reagents and Tools Table "qPCR primer of exon 10", the reverse primer (CTGGAGGACAATGACTATGG) is wrong. This primer should be reverse complement: CCATAGTCATTGTCCTCCAG. And, the primers should be named as "RT-PCR primers for detection of alternative splicing of exon 10".

3. Figure 6D, RT-PCR primers for GAPDH were not provided in the Reagents and Tools Table.

Referee #3:

The authors addressed all of my points adequately and improved the quality of their manuscript significantly. Hence I would like to recommend the revised version of "Transcriptome-wide decoding the roles of aberrant splicing in melanoma MAPK-targeted resistance evolution" for publication.

Dear reviewers,

Thank you for reviewing our manuscript entitled "*Transcriptome-wide decoding the roles of aberrant splicing in melanoma MAPK-targeted resistance evolution*". Thank you for your constructive feedback and the positive editorial decision. We are pleased to confirm that all requested revisions have been carefully addressed as follows:

Editorial Requests:

- Reordered sections
- Split EV tables (now Dataset EV1-2 & Table EV1-5)
- Updated Methods/Appendix/Reagents files

Figure Legends Revisions:

- Added exact p-values and statistical tests
- Defined box plots (median, IQR, whiskers at $1.5 \times \text{IQR}$, sample sizes)
- Clarified statistical test, error bars and sample sizes

Adopted present tense for the abstract as suggested.

All changes are implemented in the manuscript and supplementary files. Please let us know if further adjustments are needed.

We would also like to convey our gratitude to the reviewers for their diligent review of our manuscript and providing comments and suggestions to further improve the quality of our paper. We have carefully considered each comment and provided our point-by-point response below.

Response to Referee #1:

The authors have adequately addressed the concerns raised. However, the delta-PSI values presented on the y-axis of Response Fig. 4A appear modest, suggesting that alternative splicing changes in AKT2 alone may not fully explain the observed cellular phenotype. As such, we recommend that any claims regarding the role of AKT2 splicing be appropriately tempered in the manuscript.

Response: We appreciate the reviewer's insightful comment. We fully agree that the biological interpretation should reflect the quantitative evidence. In response, we have

made modifications to the manuscript.

Accordingly, we have made additional descriptions on pages 12, lines 224-226:

“These observations led us to hypothesize that alternative splicing of AKT2 may represent an additional mechanism contributing to MAPKi resistance, akin to other members of the AKT family.”

And on pages 13, lines 237-239: “This reciprocal splicing switch may explain why no significant change was observed at the gene expression level of AKT2 between baseline and disease-progressive biopsies in our previous work (Hugo et al., 2015).”

And on pages 13, lines 246-250: “Consistent with this observation, we found that there was a significant correlation between the log₂FC value of AKT2-206 and the Delta-Percent-Spliced-In (PSI) value of exon 10 (R = 0.41, p = 0.038, Appendix Fig. S9B), indicating that changes in AKT2-206 expression may be associated with the splicing regulation of exon 10.”

Response to Referee #2:

In this revision, the authors have adequately addressed all of my questions. The conclusions of this study have been strengthened, and the revised manuscript has been greatly improved. However, there are still minor points that the authors need to address.

Response: We sincerely appreciate the reviewers' recognition of the revisions made to our manuscript.

Minor points:

1. Figure 6D shows images of RT-PCR products run on agarose gels, but the authors did not provide the RT-PCR method in the Materials and Methods section.

Response: We sincerely appreciate the reviewer's careful reading and constructive comments. We apologize for this oversight. As suggested, we have now included the detailed RT-PCR protocol in the Methods section.

Accordingly, we have made additional descriptions on pages 29-30, lines 569-579:

“Total RNA was isolated using TRIzol reagent (Invitrogen). cDNA was prepared using HiScript III RT SuperMix for quantitative real-time PCR (qRT-PCR) following the manufacturer’s instructions. qRT-PCR was performed on the CFX96 Real-Time PCR Machine (Bio-Rad). Reactions were performed in 20 µl volumes containing 1x SYBR Green Supermix and 0.4 µM each of the forward and reverse primers. In qPCR thermal cycling, cDNA undergoes denaturation at 95°C, primer annealing at 60°C, and extension at 72°C by DNA polymerase. Fluorescence is measured during each extension phase as the dye incorporates into newly synthesized DNA. This cyclic process typically repeats for 40 cycles, with real-time fluorescence data collection for quantification. PCR products were identified by 2% agarose gel electrophoresis and ethidium bromide visualization.”

2.Reagents and Tools Table "qPCR primer of exon 10", the reverse primer (CTGGAGGACAATGACTATGG) is wrong. This primer should be reverse complement: CCATAGTCATTGTCCTCCAG. And, the primers should be named as "RT-PCR primers for detection of alternative splicing of exon 10".

Response: We appreciate the reviewer's meticulous review and valuable corrections. We sincerely apologize for these oversights and have now implemented the following revisions in the Reagents and Tools Table.

Oligonucleotides and other sequence-based reagents		
RT-PCR primers for detection of alternative splicing of exon 10	This paper	F: CTCGGCTCTTGAGTACTTG R: CCATAGTCATTGTCCTCCAG

3.Figure 6D, RT-PCR primers for GAPDH were not provided in the Reagents and Tools Table.

Response: We greatly appreciate the reviewer's attentive observation. We sincerely apologize for this omission and have now supplemented the complete primer information for GAPDH in the Reagents and Tools Table as follows:

Oligonucleotides and other sequence-based reagents		
RT-PCR primer of GAPDH	This paper	F:GATTCCACCCATGGCAAATTC R:CTGGAAGATGGTGTGATGGGATT

Response to Referee #3:

The authors addressed all of my points adequately and improved the quality of their manuscript significantly.

Hence I would like to recommend the revised version of "Transcriptome-wide decoding the roles of aberrant splicing in melanoma MAPK-targeted resistance evolution" for publication.

Response: We appreciate the reviewers' recognition of the revisions made to our manuscript, as well as their approval of its suitability for publication in *EMBO Reports*.

Prof. Hubing Shi
Sichuan University
17, 3rd Section, Renmin South Road
Sichuan 610041
China

Dear Prof. Shi,

I am very pleased to accept your manuscript for publication in the next available issue of EMBO reports. Thank you for your contribution to our journal.

Yours sincerely,
